# Selective Response Strategies for GenAI

**Boaz Taitler** [1]    **Omer Ben-Porat** [1]

## Abstract

The rise of Generative AI (GenAI) has significantly impacted human-based forums like Stack Overflow, which are essential for generating high-quality data. This creates a negative feedback loop, hindering the development of GenAI systems, which rely on such data to provide accurate responses. In this paper, we provide a possible remedy: A novel strategy we call *selective response*. Selective response implies that GenAI could strategically provide inaccurate (or conservative) responses to queries involving emerging topics and novel technologies, thereby driving users to use human-based forums. We show that selective response can potentially have a compounding effect on the data generation process, increasing both GenAI's revenue and user welfare in the long term. From an algorithmic perspective, we propose an approximately optimal approach to maximize GenAI's revenue under social welfare constraints. From a regulatory perspective, we derive sufficient and necessary conditions for selective response to improve welfare.

## 1. Introduction

The maxim, "Better to remain silent and be thought a fool than to speak and to remove all doubt," offers a compelling perspective on the strategic value of withholding information. While often invoked in interpersonal contexts, it resonates surprisingly well in the context of Generative AI (GenAI) systems like ChatGPT. These systems are designed to answer user queries immediately, yet one might wonder: Are there situations where the system should remain silent?

One such scenario arises when the system hallucinates. Hallucinations, defined as the generation of incorrect or fabricated information, are an intrinsic property of generative models that cannot be entirely avoided (Kalai & Vempala,

[1]Technion—Israel Institute of Technology, Israel. Correspondence to: Boaz Taitler <boaztaitler@campus.technion.ac.il>.

*Proceedings of the $42^{nd}$ International Conference on Machine Learning*, Vancouver, Canada. PMLR 267, 2025. Copyright 2025 by the author(s).

2024). Another scenario involves questions concerning safety and ethics, with potentially life-threatening consequences (Shin, 2023; Mello & Guha, 2023; Li et al., 2024). However, as we argue in this paper, it can be advantageous for both GenAI operators and users if the system avoids responding indiscriminately to every prompt, especially when addressing emerging technologies and novel content.

To illustrate, consider GenAI's competitive relationship with a human-driven platform like Stack Overflow. Users may direct their questions to either GenAI or Stack Overflow, seeking solutions to their problems. Posting a code-related question on Stack Overflow generates clarification questions in the comments, solutions offered by experts, feedback from other users (upvotes) and the original poster (acceptance flag), etc. Such valuable data could significantly enhance GenAI, improving its performance. In contrast, querying GenAI can lead to quicker user satisfaction and increased engagement with GenAI, potentially enhancing its revenue streams. On the downside, the lack of community interaction may result in less comprehensive solutions and reduce the opportunity for generating rich, labeled data that community-driven platforms like Stack Overflow thrive on (del Rio-Chanona et al., 2024; Burtch et al., 2024; Li & Kim, 2024). This absence of dynamic, user-generated content and in-depth discussions can be detrimental to user welfare in the long term, as GenAI's ability to provide high-quality answers depends on such data.

Motivated by the issue above, this paper pioneers the framework of *selective response*. Namely, strategically choosing when, if, and how to engage with user queries, particularly those involving emerging topics and novel technologies. We explicitly suggest that when a new topic emerges, GenAI could strategically decide to provide lower-quality answers than what it can or even disclaim to have not enough data to respond. We represent such behavior abstractly by modeling GenAI as not responding or "remaining silent". Clearly, selective response has a short-term negative impact; however, as we show, an appropriate selective response would lead to an improved data generation process, benefiting the long term for both GenAI's revenue and user social welfare.

**Our contribution**    Our contribution is two-fold. The first is conceptual: Our paper is the first to explore selective response for GenAI. We present a stylized model of an ecosys-

tem that evolves sequentially, featuring two platforms: A generative AI-based platform called GenAI and a human-driven Q&A platform named Forum. GenAI generates revenue by engaging with users and can adopt a *selective response strategy*: Determining the proportion of users it responds to in each round. Here, "not responding" represents a broad spectrum of possible behaviors—such as strategically withholding data, providing lower-quality answers than GenAI can produce, or claiming insufficient data, ultimately driving users to seek answers on Forum.[1] We treat these behaviors collectively as "selective response," which abstracts them for conceptual clarity. In contrast, Forum operates as a non-strategic player.

Users decide between GenAI and Forum based on the utility they derive from each platform. Those who choose Forum contribute to the creation of new data, which GenAI can later incorporate during retraining. Crucially, GenAI's quality in each round depends on the cumulative data generated since the beginning of the interaction. Our novel model allows us to explore the dynamics of content creation, welfare, and revenue from a game-theoretic lens.

Our second contribution is technical: We begin by demonstrating that selective response can Pareto-dominate the always-responding approach. Specifically, we establish the following result.

**Theorem 1.1** (Informal statement of Observation 3.1). *Compared to the case where GenAI always answers, selective response strategies can improve user welfare, GenAI's revenue, and even both.*

We also quantify the extent to which selective response can improve revenue and welfare w.r.t. the always-responding approach.

Next, we analyze the long-term effects of selective response, revealing that it leads to higher proportions of users choosing GenAI and increased data generation (Theorem 4.1). Building on this result, we devise an approximately optimal solution to GenAI's revenue maximization problem.

**Theorem 1.2** (Informal statement of Theorem 4.4). *Let $\varepsilon$ be a small positive constant and let $A$ be a finite set of selective responses. There exists an algorithm guaranteeing an additive $O(\varepsilon T^2)$ approximation of GenAI's optimal revenue, and its runtime is $O\left(\frac{T^2|A|}{\varepsilon}\right)$.*

We extend this result to the case where GenAI is constrained to meet an exogenously given social welfare threshold.

Finally, we analyze the impact of selective response on social welfare. We provide valuable insights into how a

one-round intervention affects the data generation process and its implications on welfare. We leverage these insights to demonstrate how regulators that aim to enhance social welfare can have successful one-round interventions, improving user welfare while ensuring a bounded impact on GenAI's revenue.

Altogether, our work challenges the conventional notion that GenAI should always provide answers. Despite its theoretical nature, the messages our paper conveys can translate into practical considerations for both GenAI companies and regulators and influence how forum-GenAI collaborations should form.

### 1.1. Related Work

The literature on generative AI is growing at an immense pace. Most research focuses on mitigating hallucinations (Ji et al., 2023), performance (Frieder et al., 2024; Kocoń et al., 2023; junyou li et al., 2024; Chow et al., 2025), and expanding applications (Kasneci et al., 2023; Liu et al., 2024). Our work connects to the emerging body of research on foundation models and game theory (Raghavan, 2024; Laufer et al., 2024; Conitzer et al., 2024; Dean et al., 2024). This literature studies competition between generative AI models and human content creators (Yao et al., 2024; Esmaeili et al., 2024; Keinan & Ben-Porat, 2025), the impact of generative AI on content diversity (Raghavan, 2024), and works motivated by social choice and mechanism design (Conitzer et al., 2024; Sun et al., 2024).

The most closely related work to ours is that of Taitler & Ben-Porat (2025), which examines whether the existence of generative AI is beneficial to users. In their model, the generative AI platform decides when to train, and they propose a regulatory approach to ensure social welfare for users. In contrast, our model introduces a different approach, where the generative AI chooses a portion of queries to answer, demonstrating that responding selectively can benefit both the generative AI platform and its users.

Our notion of selective response is also inspired by the economic literature on information design (Bergemann & Morris, 2019; Bergemann et al., 2015), which explores how the strategic disclosure and withholding of information can influence agents' behavior within a system. Another related concept is signaling (Crawford & Sobel, 1982; Milgrom, 1981), referring to strategic communication used by agents to potentially improve outcomes (Babichenko et al., 2024; Lu et al., 2023). Similarly, cheap talk (Lo et al., 2023; Crandall et al., 2018) can be used for fostering cooperation. In that sense, selective response can be observed as an information design problem, where GenAI strategically manages information disclosure to influence user behavior and ultimately optimize its revenue. Also related is the strand of literature of algorithmic deferring (Hemmer et al., 2023;

---

[1]In real-world scenarios, multiple GenAI systems vie for user traffic, making the analysis of such competition significantly more complex. We address this complexity in Section 7.

Mozannar & Sontag, 2020), where the algorithm can defer questions and tasks to other experts.

Finally, since our model includes an ecosystem with two platforms (GenAI and Forum), it relates to a growing body of work on competition between platforms (Rietveld & Schilling, 2021; Karle et al., 2020; Bergemann & Bonatti, 2024; Tullock, 1980; McIntyre & Srinivasan, 2017). Previous works explore the effects of competition in marketplaces on users' social welfare (Jagadeesan et al., 2023; Feldman et al., 2013), as we do in this paper.

## 2. Model

We consider a sequential setting over $T$ discrete rounds, where in each round, users interact either with Generative AI (GenAI) or a complementary human-driven platform, Forum. An instance of our problem is represented by the tuple $\langle a, \gamma, r, \beta, w^s \rangle$, and we now elaborate on the components of the model.

**GenAI.** GenAI adopts a selective response strategy $\mathbf{x} = (x_1, x_2, \ldots, x_T)$, where $x_t \in [0, 1]$ represents the proportion of users who receive answers in round $t$ among those who have already chosen GenAI. For example, $x_t = 1$ means that GenAI answers all users who selected it in round $t$, whereas $x_t = 0$ means it answers none. The performance of GenAI depends on the cumulative amount of data it has collected and trained on at the start of each round $t$, denoted $\mathcal{D}_t(\mathbf{x})$. The quality of GenAI is represented by the *accuracy function* $a(\mathcal{D}_t(\mathbf{x}))$, a strictly increasing function $a : [0, T] \to [0, 1]$, satisfying $\frac{da(\mathcal{D})}{d\mathcal{D}} > 0$ for all $\mathcal{D} \in \mathbb{R}_{\geq 0}$.[2]

We use superscripts $g$ and $s$ to denote the utility users receive from GenAI and Forum, respectively. The (expected) utility users derive from GenAI in round $t$, denoted $w_t^g(\mathbf{x})$, reflects the average quality $a(\mathcal{D}_t(\mathbf{x}))$ that users obtain from GenAI. It is given by

$$w_t^g(\mathbf{x}) = a(\mathcal{D}_t(\mathbf{x})) \cdot x_t. \tag{1}$$

Crucially, GenAI can intentionally respond less accurately than its maximum capability. In each round $t$, the proportion of users who choose GenAI is denoted by $p_t(\mathbf{x})$. This fraction is determined by the selective response strategy $\mathbf{x}$ and user decisions, which will be discussed shortly.

The (time-discounted) revenue of GenAI over $T$ rounds, $U(\mathbf{x})$, is defined by

$$U(\mathbf{x}) = \sum_{t=1}^{T} \gamma^t r(p_t(\mathbf{x})),$$

---

[2]We use the term accuracy for simplicity, allowing us to address user satisfaction abstractly. Evaluating the performance of GenAI is significantly more complex.

where $\gamma^t$ represents the discount factor applied to the revenue at round $t$, reflecting the decreasing value of future revenue. The function $r : [0, 1] \to \mathbb{R}_{\geq 0}$ maps the proportion of users $p_t(\mathbf{x})$ in round $t$ to revenue, and is assumed to be both non-decreasing and $L_r$-Lipschitz. For instance, a superlinear $r$ captures the compounding market effects of GenAI, where revenue grows at an accelerating rate as the proportion of users increases (Katz & Shapiro, 1985; Bailey et al., 2022; McIntyre & Srinivasan, 2017). Indeed, this is the case if a higher user base attracts disproportionately more offers for collaborations and investment opportunities (rich getting richer).

**Data Accumulation.** The cumulative data available to GenAI evolves as users interact with Forum. At the start of round $t$, the cumulative data $\mathcal{D}_t(\mathbf{x})$ is defined recursively as:

$$\mathcal{D}_t(\mathbf{x}) = \mathcal{D}_{t-1}(\mathbf{x}) + (1 - p_{t-1}(\mathbf{x})),$$

with the initial condition $\mathcal{D}_1(\mathbf{x}) = 0$. This initial condition represents the emergence of a new topic, where GenAI has not acquired any relevant data from previous training sets.

**Forum.** Forum provides a human-driven platform where users can post and answer questions. The utility users derive from Forum, $w^s$, is constant across rounds and satisfies $w^s \in [0, 1]$.

**Users.** Users decide between GenAI and Forum by comparing the expected utility they derive from each platform. We model user decisions using a softmax function:

$$\sigma_t(\mathbf{x}) = \frac{e^{\beta w_t^g(\mathbf{x})}}{e^{\beta w_t^g(\mathbf{x})} + e^{\beta w^s}},$$

where $\beta > 0$ is a sensitivity parameter that captures users' responsiveness to utility differences.

Recall that $x_t$ represents the proportion of users in $\sigma_t(\mathbf{x})$ who receive an answer from GenAI. The remaining users, who do not receive an answer, can either post their question on Forum or leave them unanswered. We assume the former, meaning that $p_t(\mathbf{x}) = x_t \sigma_t(\mathbf{x})$ is the proportion of users who receive an answer from GenAI, while the rest contribute to data generation by posting their question on Forum.

**User Welfare.** The *instantaneous user welfare* $w_t(\mathbf{x})$ accounts for the utilities derived from both platforms in round $t$. It is defined by

$$w_t(\mathbf{x}) = p_t(\mathbf{x}) \cdot a(\mathcal{D}_t(\mathbf{x})) + (1 - p_t(\mathbf{x}))w^s. \tag{2}$$

The *cumulative user welfare*, $W$ is therefore the sum of the instantaneous welfare over all the rounds $W(\mathbf{x}) = \sum_{t=1}^{T} w_t(\mathbf{x})$.

**Assumptions and Useful Notations** As we explain later, the following assumption on the structure of the accuracy function is crucial for analyzing the dynamics of the data generation process.

**Assumption 2.1.** The accuracy function $a(\mathcal{D})$ is $L_a$-Lipschitz with constant $L_a \leq \frac{4}{\beta}$.

We further discuss this assumption in Section 7. Additionally, we use the following notions throughout the paper. Given an arbitrary strategy $\mathbf{x}$, any strategy $\mathbf{x}^\tau$ that is obtained by reducing the response level in round $\tau$ and maintaining the other entries of $\mathbf{x}$ is called a $\tau$-*selective modification of* $\mathbf{x}$. That is, $\mathbf{x}^\tau$ is any strategy that is identical to strategy $\mathbf{x}$ except for round $\tau$, in which it *answers less than* $x_\tau$. Formally, $x_\tau^\tau \in [0, x_\tau)$ and $x_t^\tau = x_t$ for every $t \neq \tau$. For brevity, if $\mathbf{x}$ is clear for the context, we use $\mathbf{x}^\tau$ as any arbitrary $\tau$-selective modification. Another useful notation in $\bar{\mathbf{x}}$, where $\bar{\mathbf{x}} = (1, 1, \dots, 1)$ is *full response* or the *always-responding* strategy; we use these interchangeably. We use this strategy as a point of comparison, establishing a baseline to test other strategies.

**Example 2.2.** Consider the instance $T = 10$, $a(\mathcal{D}) = 1 - e^{-0.3\mathcal{D}}$, $\gamma = 0.9$, $r(p) = p^2$, $\beta = 10$, and $w^s = 0.5$. Consider the following selective response strategy $\bar{\mathbf{x}} = (1, \dots, 1)$ and $\mathbf{x}$ which is defined by.

$$x_t = \begin{cases} 0 & t \leq 4 \\ 1 & \text{otherwise} \end{cases}.$$

At $t = 1$ it holds that $d_1(\mathbf{x}) = d_1(\bar{\mathbf{x}}) = 0$. Notice that $a(0) = 0$ and therefore $p_1(\bar{\mathbf{x}}) = 1 \cdot \frac{1}{1+e^{\beta w^s}} \approx 0.0067$. Similarly, for $\mathbf{x}$ it is $p_1(\mathbf{x}) = 0 \cdot \frac{1}{1+e^{\beta w^s}} = 0$. Thus, the generated data is $d_2(\bar{\mathbf{x}}) \approx 1 - 0.0067 = 0.9933$ and $d_2(\mathbf{x}) = 1$.

With that, we have the ingredients to calculate the instantaneous welfare at time $t = 1$.

$$\begin{aligned} w_1(\bar{\mathbf{x}}) &= p_1(\bar{\mathbf{x}})a(\mathcal{D}_t(\bar{\mathbf{x}})) + (1 - p_1(\bar{\mathbf{x}}))w^s \\ &\approx 0.0067 \cdot 0 + 0.9933 \cdot 0.5 \approx 0.4966, \\ w_1(\mathbf{x}) &= 0 \cdot 0 + 1 \cdot w^s = 0.5. \end{aligned}$$

Figure 1 demonstrates the proportions of the strategies $\bar{\mathbf{x}}$ and $\mathbf{x}$ as a function of the round for $t \in [T]$. Notice that the selective response $\mathbf{x}$ induces lower user proportions in the earlier rounds, but it eventually surpasses the full response strategy $\bar{\mathbf{x}}$.

Finally, the revenue is attained by calculating $U(\mathbf{x}) = \sum_{t=1}^{10} \gamma^{t-1}(p_t(\mathbf{x}))^2$. Computing this for the two strategies, we see that $U(\mathbf{x})$ is roughly 5% higher than $U(\bar{\mathbf{x}})$. Similarly, the welfare $W(\mathbf{x})$ is about 7.6% higher than $W(\bar{\mathbf{x}})$. As this example suggests, selective response can improve both revenue and welfare. Indeed, this is the focus of the next section.

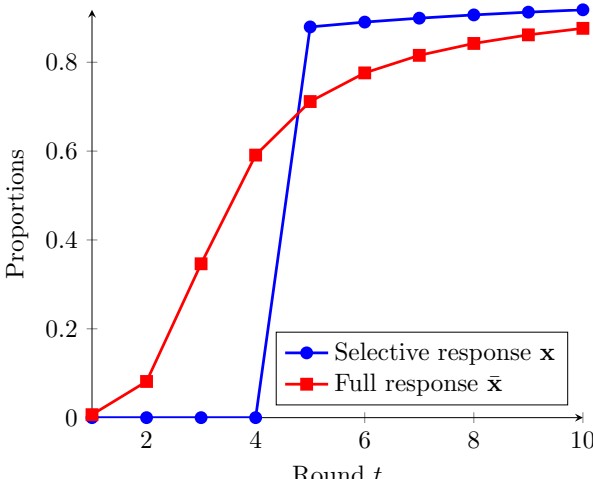

Figure 1: A visualization for Example 2.2. The blue (circle) curve shows the proportion of users $p_t(\mathbf{x})$ for the selective response strategy $\mathbf{x}$ at each round $t$. The red (square) curve depicts the corresponding proportion for the full response.

## 3. The Benefits of Selective Response

This section motivates our work by showing that selective response may benefit both GenAI and its users. We first demonstrate a qualitative result: Selective response can improve revenue, welfare, or both. Then, in Subsection 3.1, we quantify the extent of these improvements.

Recall the definition of the full response strategy $\bar{\mathbf{x}}$. We use it as a benchmark in evaluating the potential impact of adopting a selective response strategy on GenAI and its users.

**Observation 3.1.** *There exist instances and a selective response strategy* $\mathbf{x}$ *that satisfy each one of the following inequalities:*

1. $U(\mathbf{x}) > U(\bar{\mathbf{x}})$ *and* $W(\mathbf{x}) > W(\bar{\mathbf{x}})$,

2. $U(\mathbf{x}) < U(\bar{\mathbf{x}})$ *and* $W(\mathbf{x}) > W(\bar{\mathbf{x}})$,

3. $U(\mathbf{x}) > U(\bar{\mathbf{x}})$ *and* $W(\mathbf{x}) < W(\bar{\mathbf{x}})$.

The first inequality in Observation 3.1 indicates that there exists a selective response strategy that Pareto dominates the always-responding strategy. The subsequent two inequalities imply that increasing either GenAI's revenue or the users' social welfare may come at the expense of the other.

### 3.1. Price of Always Responding

In this subsection, we quantify the negative impact of always answering users' queries. We introduce two indices: RPAR, an abbreviation for **R**evenue's **P**rice of **A**lways

**R**esponse, and WPAR, which stand for **W**elfare's **P**rice of **A**lways **R**esponse. Formally, RPAR $\triangleq \frac{\max_{\mathbf{x}} U(\mathbf{x})}{U(\bar{\mathbf{x}})}$ and WPAR $\triangleq \frac{\max_{\mathbf{x}} W(\mathbf{x})}{W(\bar{\mathbf{x}})}$ are the price of always answering with respect to revenue and social welfare, respectively. These metrics capture the inefficiencies in revenue and welfare that arise when GenAI always responds to all user queries. Our next result demonstrates that the revenue inefficiency is unbounded.

**Proposition 3.2.** *For every $M \in \mathbb{R}_{>0}$ there exists an instance I with $L_r = \Theta(\ln(M))$ such that $\mathrm{RPAR}(I) > M$.*

Proposition 3.2 relies on the revenue scaling function $r(p)$, which can bias GenAI's incentives toward data generation rather than immediate revenue. For example, when $r(p)$ takes the form of a sigmoid function, the parameter $L_r$ controls the steepness of the curve. If the sigmoid is sufficiently steep, $r(p)$ approximates a step function, requiring GenAI to surpass a certain user proportion threshold to generate revenue. This mirrors threshold-based incentives, where substantial rewards are only provided once a predefined threshold is met.

Our next proposition shows that there exist instances where selective responses can result in social welfare nearly twice as large as that of the always-responding strategy.

**Proposition 3.3.** *For every $\varepsilon > 0$ there exists an instance I with $\mathrm{WPAR}(I) > 2 - \varepsilon$.*

We end this section by analyzing Price-of-Anarchy (Koutsoupias & Papadimitriou, 1999; Roughgarden, 2005), a standard economic concept that measures the harm due to strategic behavior of GenAI. Formally, $PoA = \frac{\max_{\mathbf{x}} W(\mathbf{x})}{\min_{\mathbf{x} \in \mathcal{R}} W(\mathbf{x})}$, where $\mathcal{R}$ is the set of revenue-maximizing strategies. We show that it can increase with the smoothness parameter of the reward function $L_r$. Since this analysis depends on the revenue-optimal strategy of GenAI, which we only examine in later sections, we defer this analysis to the Appendix B.1.

# 4. The Impact of Selective Response on GenAI's Revenue

In this section, we analyze the revenue-maximization problem faced by GenAI. Subsection 4.1 examines the impact of using selective responses on both user proportions and generated data. We show that any $\tau$-selective modification of any strategy and any $\tau$ generates more future data and attracts more users to GenAI from round $\tau + 1$ onward. Subsequently, we develop two approaches for maximizing GenAI's welfare. In Subsection 4.2, we develop an approximately optimal algorithm for maximizing GenAI's optimal revenue. In Subsection 4.3, we focus on undiscounted settings, i.e., $\gamma = 1$, and consider a welfare-constraint revenue maximization: Maximizing revenue under a minimal social welfare level constraint. We emphasize the trade-off

between our approaches: The first approach cannot handle welfare constraints, while the second is restricted to undiscounted revenue.

## 4.1. Selective Response Implies Increased User Proportions

Next, we analyze the impact of using a $\tau$-selective modification of any base strategy on the proportions and data generation. At first glance, using selective responses harms immediate revenue, as it encourages users to turn to Forum. However, as suggested by Observation 3.1, lower response levels can ultimately prove beneficial. But why is this the case?

The answer lies in the dynamics of data generation. By employing a more selective response, GenAI incentivizes users to engage with Forum, which results in the creation of more data. This additional data becomes crucial in future rounds, enabling GenAI to attract a more significant user proportion in subsequent interactions. While this reasoning is intuitive, its application over time presents a technical challenge: As the proportion of users choosing GenAI increases, the marginal data generated per round may decrease, potentially leading to less data than a strategy where GenAI answers every query. However, the theorem below demonstrates the compounding effect of selective response, guaranteeing consistently higher user proportions in future rounds.

**Theorem 4.1.** *Fix any strategy $\mathbf{x}$. For every $t > \tau$ it holds $\mathcal{D}_t(\mathbf{x}^\tau) > \mathcal{D}_t(\mathbf{x})$ and $p_t(\mathbf{x}^\tau) \geq p_t(\mathbf{x})$ where $p_t(\mathbf{x}^\tau) = p_t(\mathbf{x})$ if and only if $x_t = x_t^\tau = 0$.*

**Proof sketch of Theorem 4.1.** To prove this theorem, we first show that $\mathcal{D}_t(\mathbf{x}^\tau) - \mathcal{D}_t(\mathbf{x}) > 0$ for every $t > \tau$. To do so, we introduce some additional notations. First, we define $Q(\mathcal{D}, x) = x \frac{e^{\beta a(\mathcal{D})x}}{e^{\beta a(\mathcal{D})x} + e^{\beta w^s}}$, which represents the resulting proportion when using a selective response $x$ with data $\mathcal{D}$. Next, we define $f(\mathcal{D}, x) = D - Q(\mathcal{D}, x)$ as the total data generated when choosing $x$ with initial data $D$. Note that for every $t \in [T]$, we have $f(\mathcal{D}_t(\mathbf{x}), x_t) = \mathcal{D}_{t+1}(\mathbf{x})$ and $Q(\mathcal{D}_t(\mathbf{x}), x_t) = p_t(\mathbf{x})$. Following, we prove that $f(\mathcal{D}, x)$ is monotonically increasing with respect to $\mathcal{D}$.

**Proposition 4.2.** *For every $x \in [0, 1]$ and $\mathcal{D} \in \mathbb{R}_{\geq 0}$ it holds that $\frac{df(\mathcal{D}, x)}{d\mathcal{D}} > 0$.*

Proposition 4.2, combined with Assumption 2.1, imply that for every $t > \tau$, if $\mathcal{D}_t(\mathbf{x}^\tau) > \mathcal{D}_t(\mathbf{x})$, then it follows that $\mathcal{D}_{t+1}(\mathbf{x}^\tau) > \mathcal{D}_{t+1}(\mathbf{x})$. Iterating Proposition 4.2 leads to $\mathcal{D}_t(\mathbf{x}^\tau) > \mathcal{D}_t(\mathbf{x})$ for every $t > \tau$. Finally, since $Q(\mathcal{D}, x)$ is monotonically increasing with respect to $\mathcal{D}$, we conclude that $p_t(\mathbf{x}^\tau) \geq p_t(\mathbf{x})$ for every $t > \tau$, thus completing the proof of Theorem 4.1. $\square$

## 4.2. Revenue Maximization

In this subsection, we develop an approximately optimal algorithm for maximizing GenAI's revenue. We begin by noting the challenges of the problem, emphasizing why identifying the optimal strategy is nontrivial.

Recall that Theorem 4.1 demonstrates that employing a selective response increases future proportions. This argument can be applied iteratively by employing selective responses in different rounds, further enhancing the future proportions. This intuition hints that a step function-based strategy could be optimal: GenAI should answer no queries in early rounds and then answer all queries. In such a case, the effective space of optimal strategy reduces to all $T$ step function-based strategies. Unfortunately, this intuition is misleading.

**Observation 4.3.** *There exist instances where the optimal strategy* $\mathbf{x}^\star \notin \{0,1\}^T$.

Due to Observation 4.3, the search for optimal strategies spans the continuous domain $[0,1]^T$. This observation motivates us to adopt an approximation-based approach to identify near-optimal strategies efficiently. To that end, we devise the ASR algorithm, which stands for **A**pproximately optimal **S**elective **R**esponse. ASR follows a standard dynamic programming structure, but its approximation analysis is nontrivial, as we elaborate below. Therefore, we introduce it in Appendix C.2 and provide an informal description here, along with key insights from its analysis.

**Overview of the** ASR **algorithm**  Fix any finite set $A$, $A \subset [0,1]$. Naively, if we wish to find $\arg\max_{\mathbf{x} \in A^T} U(\mathbf{x})$, we could exhaustively search along all $A^T$ strategies via inefficient dynamic programming. However, we show how to design a small-size state representation and execute dynamic programming effectively. The challenge is ensuring that any strategy's revenue within the small state representation approximates the actual revenue of that strategy. To achieve this, we discretize the amount of data $\mathcal{D}$. Recall that in each round, the amount of generated data is at most 1, meaning that for any strategy $\mathbf{x}$, the total data up to round $t$ is $\mathcal{D}_t(\mathbf{x}) \in [0, t-1]$. Consequently, we define states by the round $t$ and the discretized data value within $[0, t-1]$. At the heart of our dynamic programming approach is the calculation of the expected revenue for each state and action $y \in A$, based on the induced proportions, generated data, and the anticipated next state. The next theorem provides the guarantees of ASR.

**Theorem 4.4.** *Fix any instance and let $\varepsilon > 0$. The* ASR *algorithm outputs a strategy $\mathbf{x}$ such that*

$$U(\mathbf{x}) > \max_{\mathbf{x}' \in A^T} U(\mathbf{x}') - \varepsilon L_r T^2, \qquad (3)$$

*and its run time is $O\left(\frac{T^2 |A|}{\varepsilon}\right)$.*

**Proof sketch of Theorem 4.4.** To prove the theorem, there are two key elements we need to establish. First, for any two similar data quantities under any selective response strategy, the resulting revenues are similar as well. Imagine that $\mathcal{D}^1$ is the actual data quantity generated by some strategy up to some arbitrary round, and $\mathcal{D}^2$ is the discretized data quantity of the same strategy in our succinct representation. If GenAI plays $x \in A$ in the next round, how different do we expect the data quantity to be in the next round? In other words, we need to bound the difference $\left|f(\mathcal{D}^1, x) - f(\mathcal{D}^2, x)\right|$, where $f$ follows the definition from the proof of Theorem 4.1. To that end, we prove the following lemma.

**Lemma 4.5.** *For any $\mathcal{D}^1, \mathcal{D}^2 \in \mathbb{R}_{\geq 0}$ and $x \in A$, it holds that $\left|f(\mathcal{D}^1, x) - f(\mathcal{D}^2, x)\right| \leq \left|\mathcal{D}^1 - \mathcal{D}^2\right|$.*

We further leverage this lemma in proving the second key element: The discrepancy of the induced proportions is bounded by the discrepancy in the data quantities, i.e., $\left|Q(\mathcal{D}^1, x) - Q(\mathcal{D}^2, x)\right| < \left|\mathcal{D}^1 - \mathcal{D}^2\right|$. Equipped with Lemma 4.5 and the former inequality, we bound the discrepancy the dynamic programming process propagates throughout its execution. $\qquad \square$

Observe that Theorem 4.4 guarantees approximation with respect to the best strategy that chooses actions from $A$ only. Indeed, the right-hand-side of Inequality (3) includes $\max_{\mathbf{x}' \in A^T} U(\mathbf{x}')$. In fact, by taking $A$ to be the $\delta$-uniform discretization of the $[0,1]$ interval for a small enough $\delta > 0$, we can extend our approximation guarantees to the best continuous strategy at the expense of a slightly larger approximation factor.

**Theorem 4.6.** *Let $\delta \in (0, \frac{1}{\beta}]$ and let $A_\delta = \{0, \delta, 2\delta \ldots 1\}$. Let $\mathbf{x}$ be the solution of* ASR *with parameters $\varepsilon > 0$ and $A_\delta$. Then,*

$$U(\mathbf{x}) \geq \max_{\mathbf{x}'} U(\mathbf{x}') - \frac{7\beta + 1}{4(1-\gamma)^2} L_r \delta - \varepsilon L_r T^2,$$

*and the run time of* ASR *is $O\left(\frac{T^2}{\varepsilon \delta}\right)$.*

## 4.3. Welfare-Constrained Revenue Maximization

While the ASR algorithm we developed in the previous subsection guarantees approximately optimal revenue, it might harm user welfare. Indeed, Observation 3.1 implies that selective response can improve revenue but decrease welfare. This motivates the need for a welfare-constrained revenue maximization framework, where the objective is to maximize GenAI's revenue while ensuring that the social welfare remains above a predefined threshold $W$. Formally,

$$\begin{aligned} \max_{\mathbf{x} \in A^T} \ & U(\mathbf{x}) \\ \text{s.t. } & W(\mathbf{x}) \geq W. \end{aligned} \qquad (\text{P1})$$

Noticeably, if the constant $W$ is too large, that is, $W > \max_{\mathbf{x}} W(\mathbf{x})$, Problem (P1) has no feasible solutions; hence, we assume $W \leq \max_{\mathbf{x}} W(\mathbf{x})$. Our approach for this constrained optimization problem is inspired by the PARS-MDP problem (Ben-Porat et al., 2024). We reduce it to a graph search problem, where we iteratively discover the Pareto frontier of feasible revenue and welfare pairs, propagating optimal solutions of sub-problems. Due to space constraints, we defer its description to Appendix C.3 and present here its formal guarantees.

**Theorem 4.7.** *Fix an instance such that $\gamma = 1$. Let $\varepsilon > 0$ and let $\mathbf{x}^{\star}$ be the optimal solution for Problem* (P1). *There exists an algorithm with output $\mathbf{x}$ that guarantees*

1. $U(\mathbf{x}) > U(\mathbf{x}^{\star}) - \varepsilon T^2 \max\{1, L_r\}$,

2. $W(\mathbf{x}) > W - 2\varepsilon T^2 (L_a + 1)$,

*and its running time is $O\left(\frac{T^2 |A|}{\varepsilon^2} \log(\frac{T|A|}{\varepsilon})\right)$.*

Unfortunately, the technique we employed in the previous subsection for extending the approximation from the optimal discrete strategy to the optimal continuous strategy is ineffective in the constrained variant; see Section 7.

# 5. The Impact of Selective Response on Social Welfare

In this section, we flesh out the impact of implementing $\tau$-selective modifications on social welfare. Specifically, we focus on modifying an arbitrary initial strategy $\mathbf{x}$ by applying a selective response in a single round $\tau$.

The next Theorem 5.1 provides a powerful tool in characterizing the change in the instantaneous user welfare in $\tau$-selective modifications. We first present the theorem and then analyze its consequences.

**Theorem 5.1.** *Fix any instance and a strategy $\mathbf{x}$. There exist thresholds $B$ and $C$, $B \leq C < w^s$, such that for any $\tau$-selective modification $\mathbf{x}^{\tau}$ it holds that:*

1. *In round $\tau$, if $w_t^g(\mathbf{x}) < B$ then $w_{\tau}(\mathbf{x}^{\tau}) > w_{\tau}(\mathbf{x})$;*

2. *For every round $t > \tau$, if $w_t^g(\mathbf{x}^{\tau}) < B$, then it holds that $w_t(\mathbf{x}^{\tau}) < w_t(\mathbf{x})$;*

3. *For every round $t > \tau$ such that $w_t^g(\mathbf{x}) > C$, it holds that $w_t(\mathbf{x}^{\tau}) > w_t(\mathbf{x})$.*

We interpret the theorem using the illustrations in Figure 2. In Figure 2a, the horizontal axis is the round number and the vertical axis is the expected utility users obtain from GenAI, $w_t^g$. There are four curves: The red (circle) is the base strategy $\mathbf{x}$; the blue (triangle) represents a $\tau$-selective

modification $\mathbf{x}^{\tau}$; the black (dotted) line is the threshold $B$; and the orange (dashed) line is the threshold $C$. Figure 2b also uses the round number as the horizontal axis and includes both strategies $\mathbf{x}$ and $\mathbf{x}^{\tau}$, but its vertical axis shows the instantaneous welfare $w_t$.

Before round $\tau$, the strategies agree on the response levels; thus, the utilities are identical, and the blue and red curves intersect in both figures. Next, we focus on round $\tau$. Recall that the $\tau$-selective modification has a lower response level in round $\tau$, i.e., $x_{\tau}^{\tau} < x_{\tau}$. Consequently, Figure 2a demonstrates that the utility GenAI induces is lower. Part 1 of the theorem implies that the $\tau$-selective modification obtains a higher instantaneous welfare, as shown in Figure 2b. To see why, notice that both $w_{\tau}^g(\mathbf{x}^{\tau}), w_{\tau}^g(\mathbf{x})$ are less than $B < w^s$; thus, any user that is directed to Forum under the modification obtains a higher utility.

For any round $t$, $t > \tau$, the blue curve is above the red curve in Figure 2a. Namely, GenAI's quality under the $\tau$-selective modification $\mathbf{x}^{\tau}$ is greater than that of the base strategy $\mathbf{x}$. This is a direct corollary of Theorem 4.1: More data is created ($\mathcal{D}_t(\mathbf{x}^{\tau}) > \mathcal{D}_t(\mathbf{x})$) and more users choose GenAI ($p_t(\mathbf{x}^{\tau}) \geq p_t(\mathbf{x})$); hence $w_t^g(\mathbf{x}^{\tau}) > w_t^g(\mathbf{x})$.

Next, we focus on Part 2 of the theorem, which is demonstrated by the shaded gray area (featuring horizontal lines) in the two figures. In Figure 2a, the gray area represents rounds with $t > \tau$ and $w_t^g(\mathbf{x}^{\tau}) < B$. Consequently, Part 2 of Theorem 4.1 implies that the instantaneous welfare of $\mathbf{x}^{\tau}$ is lower than that of $\mathbf{x}$ (shaded area in Figure 2b).

We can reformulate the instantaneous welfare from Equation (2) to include $\sigma_t(\cdot)$, namely,

$$w_t(\mathbf{x}) = \sigma_t(\mathbf{x}) w_t^g(\mathbf{x}) + (1 - x_t \sigma_t(\mathbf{x})) w^s. \tag{4}$$

On the one hand, GenAI's expected utility increases under the $\tau$-selective modification ($w_t^g(\mathbf{x}^{\tau}) > w_t^g(\mathbf{x})$), while both utilities are under $B$ and thus under $w^s$. On the other hand, the proportion of users switching to GenAI grows: $\sigma_t(\mathbf{x}^{\tau}) > \sigma_t(\mathbf{x})$. Therefore, the first product on the right-hand side of Equation (4) increases for the $\tau$-selective modification, while the second product decreases. Part 2 quantifies this tradeoff, implying that the instantaneous welfare is decreasing overall. This is illustrated in the gray area in Figure 2b, as the red curve is above the blue curve.

Similarly, Part 3 of the theorem, represented by the green shaded area (with vertical lines), corresponds to rounds in which the red curve in Figure 2a exceeds the threshold $C$, that is, $w_t^g(\mathbf{x}) > C$. In these rounds, Part 3 asserts that the instantaneous welfare of $\mathbf{x}^{\tau}$ exceeds that of $\mathbf{x}$ (green area in Figure 2b).

Finally, we discuss the thresholds $B$ and $C$. For the latter, the theorem holds trivially for $C = w^s$. However, as we show in the proof of the theorem, we have a tighter threshold

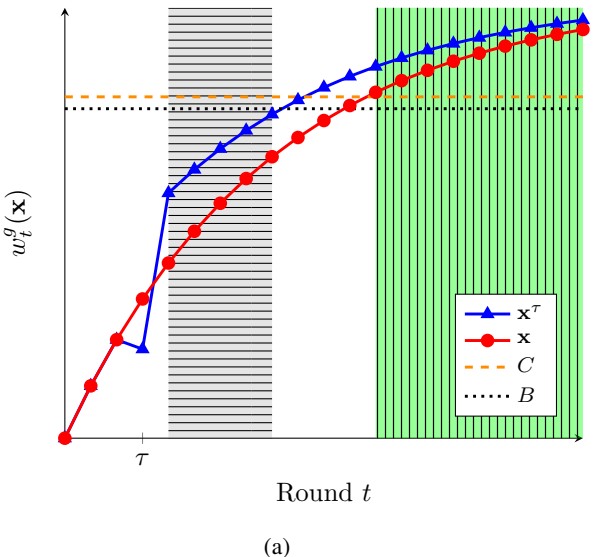
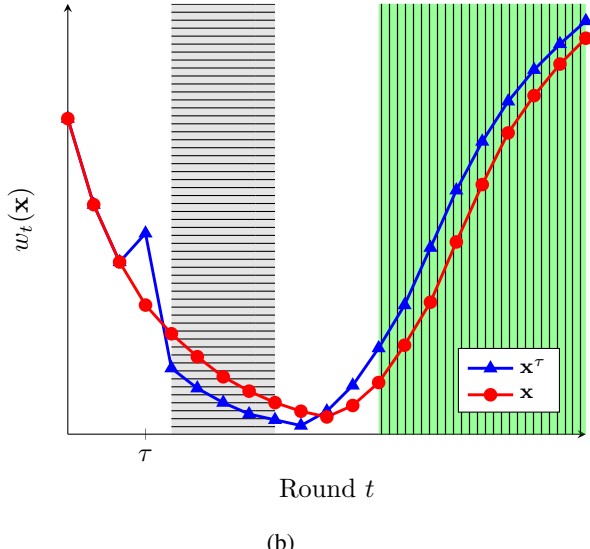

|  |
|---|
| (a) |
| (b) |

Figure 2: Illustrating Theorem 5.1. The left figure illustrates GenAI's expected utility vs round index, and the right figure illustrates instantaneous welfare vs round index. The red (circle), blue (triangle) curves represent the base strategy $\mathbf{x}$, and a $\tau$-selective modification $\mathbf{x}^\tau$. The orange (dashed) and black (dotted) lines represent the thresholds $B$ and $C$, respectively. The gray and green shaded areas highlight different parts of the theorem. The gray area indicates rounds where the condition of Part 2 is met (Figure 2a), and the resulting lower instantaneous welfare is illustrated in Figure 2b. The green region denotes rounds where $\mathbf{x}^\tau$ leads to higher instantaneous welfare (Part 3).

of $C = w^s - \frac{\mathcal{W}(e^{-1})+1}{\beta}$ where $\mathcal{W}$ is the Lambert function. As for $B$, Theorem 4.1 implies its existence, yet finding a closed-form expression remains an open question.

# 6. Regulating Selective Response for Improved Social Welfare with Minimal Intervention

In this section, we adopt the perspective of a regulator aiming to benefit users through interventions. We show how to use the results from the previous section to ensure that the intervention will be beneficial from a welfare perspective. Additionally, we bound the revenue gap that such an intervention may create. A crucial part of our approach is that the regulator can see previous actions, but not future actions, making it closer to real-world scenarios. Specifically, for any arbitrary round $\tau$, we assume the regulator observes $x_1, \ldots x_\tau$, but has no access to GenAI's future strategy $(x_t)_{t=\tau+1}^T$.

## 6.1. Sufficient Conditions for Increasing Social Welfare

We focus on $\tau$-selective modifications that guarantee to increase welfare w.r.t. a base strategy $\mathbf{x}$. We further assume GenAI commits to a 0 response level as long as its quality is below $C$, where $C$ is the threshold from Theorem 5.1. This commitment, formally given by $\min_{t>\tau}\{w_t^g(\mathbf{x}) \mid w_t^g(\mathbf{x}) > 0\} > C$, represents the minimum utility required from GenAI for rounds $t > \tau$.

**Corollary 6.1.** *Let $B$ and $C$ be the thresholds from Theorem 5.1. Assume that $w_\tau^g(\mathbf{x}) < B$ and that GenAI commits, i.e., $\min_{t>\tau}\{w_t^g(\mathbf{x}) \mid w_t^g(\mathbf{x}) > 0\} > C$ holds for all $t > \tau$. Then, $W(\mathbf{x}^\tau) \geq W(\mathbf{x})$.*

Intuitively, Corollary 6.1 ensures that the welfare improvement due to this intervention (the green shaded region in Figure 2) surpasses the welfare reduction (the gray region).

## 6.2. Bounding GenAI's Revenue Gap

A complementary question is to what extent *forcing* a $\tau$-selective response can harm GenAI's revenue. Our goal is to establish a bound on the revenue gap between the base strategy $\mathbf{x}$ and the modified strategy $\mathbf{x}^\tau$, where the selective response occurs in round $\tau$. We stress that incomplete information about future actions makes this analysis challenging.

By definition, $\mathbf{x}$ and $\mathbf{x}^\tau$ are identical except for round $\tau$. Consequently, they generate the same amount of data in all rounds *before* $\tau$. Using a $\tau$-selective response reduces the proportion of answers in that round, which in turn increases the accumulated data available in round $\tau + 1$. Therefore, the revenue gap can be decomposed into two components: (1) The immediate effect of the proportion change in round $\tau$, $r(p_\tau(\mathbf{x}^\tau)) - r(p_\tau(\mathbf{x}))$; and (2) the downstream effects on subsequent rounds due to the change in the data generation process. Using several technical lemmas that we prove in Appendix E, we show that:

**Corollary 6.2.** *It holds that*

$$U(\mathbf{x}^\tau) - U(\mathbf{x}) \leq$$

$$\gamma^{\tau-1} \left( r(p_\tau(\mathbf{x}^\tau)) - r(p_\tau(\mathbf{x})) \right) + L_r \gamma^\tau \frac{p_\tau(\mathbf{x}^\tau) - p_\tau(\mathbf{x})}{1 - \gamma}.$$

The above bound is less informative as $\gamma$ approaches 1. In Theorem E.1, we obtain a tighter bound by having some additional assumptions.

## 7. Discussion and Future Work

This paper pioneers the novel approach of selective response, showing that withholding responses can be a powerful tool for GenAI systems. By opting not to answer every query as accurately as it can—particularly when new or complex topics emerge—GenAI can encourage user participation on community-driven platforms and thereby generate more high-quality data for future training. This mechanism ultimately enhances GenAI's long-term performance and revenue. Therefore, selective response is related to active learning, nudging users to generate more data. This mirrors the exploration-exploitation tradeoff from the multi-armed bandit literature: GenAI forgoes immediate gains and risks user dissatisfaction in the pursuit of better long-term revenue. From a welfare perspective, our results indicate that selective response can benefit users, leading to better solutions and increased overall satisfaction. Since this work is the first to address selective response strategies for GenAI, numerous promising directions remain for future research; we highlight some of them below.

First, from a technical standpoint, all of the results in this paper rely on Assumption 2.1, involving the Lipshitz condition of the accuracy function and the sensitivity parameter $\beta$. Future work could seek to relax this assumption. Furthermore, our constrained optimization approach in Subsection 4.3 could be extended to approximate the optimal (continuous) strategy instead of the optimal discrete strategy.

Second, our stylized model adopts the simplifying—though unrealistic—assumption that only a single GenAI platform exists. Admittedly, this makes it easier to focus on the idea of selective responses, and indeed, this assumption is pivotal in keeping our analysis tractable. Future research could explore scenarios with multiple GenAI platforms and human-centered forums. In such settings, one platform's selective response might redirect users to competing GenAI platforms, leading to the tragedy of the commons (Hardin, 1968): Although all GenAI platforms benefit from fresh data generation, none may choose to respond selectively if it means losing users to competitors.

Third, we assumed Forum behaves non-strategically. In reality, human-centered platforms often monetize their data by selling it to GenAI platforms, adding a further layer of strategic interaction for GenAI. Moreover, data transfer between the platforms can form the basis for collaboration: GenAI could employ selective response to bolster Forum content creation, and Forum could, in turn, attribute that content to GenAI for subsequent use in retraining.

## Acknowledgments

We thank the anonymous reviewers for their helpful comments. This research was supported by the Israel Science Foundation (ISF; Grant No. 3079/24).

## Impact Statement

Our work aims to benefit both GenAI and its users by introducing a new action: allowing GenAI to choose whether to answer or to adjust the quality of its answer. As shown in this work, a wise use of selective response can be beneficial for both GenAI and its users. From an ethical perspective, using a selective response can be seen as a responsible decision, as it allows GenAI to redirect users instead of providing low-quality answers. Furthermore, if GenAI is transparent about its need for data, it can notify users about its lack of confidence and explain that it may be able to assist them better in the future by choosing not to answer in the present.

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

## A. Definitions and Notations

We first define the following function:

$$f(\mathcal{D}, x) = \mathcal{D} + x \left( 1 - \frac{e^{\beta a(\mathcal{D})x}}{e^{\beta a(\mathcal{D})x} + e^{\beta w^s}} \right).$$

Denote $q(\mathcal{D}, x) = \frac{e^{\beta a(\mathcal{D})x}}{e^{\beta a(\mathcal{D})x} + e^{\beta w^s}}$, therefore $f(\mathcal{D}, x)$ can be expressed as $f(\mathcal{D}, x) = d + (1 - xq(\mathcal{D}, x))$.

Next, we define $\lfloor \cdot \rfloor_\varepsilon$ as the discretization operator with respect to $\varepsilon \in \mathbb{R}$. Formally, for any $x \in \mathbb{R}$, the discretization operator is given by $\lfloor x \rfloor_\varepsilon = \varepsilon \lfloor \frac{x}{\varepsilon} \rfloor$.

## B. Proofs Omitted from Section 3

**Proof of Observation 3.1.** We prove each clause separately.

**1. Pareto dominance** This is shown in Example 2.2, for which it holds that

- $U(\bar{\mathbf{x}}) < 2.356$.

- $U(\mathbf{x}) > 2.483$.

- $W(\bar{\mathbf{x}}) < 5.73$.

- $W(\mathbf{x}) > 6.2$.

**2. Decreases revenue and increases welfare** Let $T = 5$ and consider the instance $a(\mathcal{D}) = 1 - e^{-0.4\mathcal{D}}$, $\gamma = 1$, $\beta = 3$, $w^s = 0.7$ and $r(p) = p$.

We calculate the revenue and the social welfare induced by $\bar{\mathbf{x}}$ by calculating the proportions for every $t \in [T]$. Therefore, the induced revenue is $U(\bar{\mathbf{x}}) > 1.6$ and the social welfare $W(\bar{\mathbf{x}}) < 3.3$.

Next, we denote $\mathbf{x} = (0, 0, \ldots, 0)$, the strategy for which GenAI never answers. By definition we have that $U(\mathbf{x}) = 0 < U(\bar{\mathbf{x}})$ and $W(\mathbf{x}) = Tw^s = 3.5 > W(\bar{\mathbf{x}})$.

**3. Increases revenue and decreases welfare** Let $T = 5$ and consider the instance $a(\mathcal{D}) = 1 - e^{-0.4\mathcal{D}}$, $\gamma = 1$, $\beta = 3$, $w^s = 0.1$ and $r(p)$ is the step function defined as

$$r(p) = \begin{cases} 1 & p \geq q(4, 1) \\ 0 & \text{Otherwise} \end{cases}.$$

We denote $\mathbf{x}$ the strategy that satisfies

$$x_t = \begin{cases} 1 & t = T \\ 0 & \text{Otherwise} \end{cases}.$$

Notice that $p_1(\bar{\mathbf{x}}) > 0$ and therefore for every $t \in [5]$ it holds that $D_t(\bar{\mathbf{x}}) < 4$. Thus, $U(\bar{\mathbf{x}}) = 0$.

The revenue induced by $\mathbf{x}$ is equal to the revenue induced at round $T$. This is true since $x_t = 0$ for every $t < T$ and therefore $p_t(\mathbf{x}) = 0$. At round $T$, the total generated data is $\mathcal{D}_T(\mathbf{x}) = T - 1 = 4$. Thus, $U(\mathbf{x}) = r(p_T(\mathbf{x})) = q(4, 1) > 0.89$

Calculating the welfare induced by $\bar{\mathbf{x}}$ can be done by calculating $p_t(\bar{\mathbf{x}})$, resulting in $W(\bar{\mathbf{x}}) > 1.17$.

Similarly, we can calculate the welfare induced by strategy $\mathbf{x}$. Repeating the same calculation leads to $W(\mathbf{x}) < 1.122$; thus, we can conclude that $U(\bar{\mathbf{x}}) < U(\mathbf{x})$ and $W(\bar{\mathbf{x}}) > W(\mathbf{x})$. This completes the proof of Observation 3.1. □

### B.1. Proofs Omitted from Subsection 3.1

**Proof of Proposition 3.2.** Consider the instance $a(\mathcal{D}) = \frac{1+\mathcal{D}}{T}$, $\gamma = 1$, $\beta = 1$, $w^s = \frac{1}{T}$ and $r(p)$ is the sigmoid function defined as $r(p) = \frac{1}{1+e^{-\xi(q(T-1,1)-p)}}$, such that $\xi = \frac{\ln(2TM)}{q(T-1,1)-q(\frac{T-1}{2},1)}$.

Notice that for every $t \in T$ it holds that $w_t^g(\bar{\mathbf{x}}) = a(\mathcal{D}_t(\bar{\mathbf{x}})) = \frac{1+\mathcal{D}_t(\bar{\mathbf{x}})}{T} > \frac{1}{T} = w^s$. Therefore, we get that $p_t(\bar{\mathbf{x}}) > 0.5$ and $\mathcal{D}_t < \frac{t-1}{2}$.

we now bound the revenue induced by $\bar{\mathbf{x}}$.

$$U(\bar{\mathbf{x}}) = \sum_{t=1}^{T} r(p_t(\mathbf{x})) < Tr(q(\frac{T-1}{2}, 1)).$$

Next, we define the scheme that answers only at the last round $\mathbf{x}^\star = (0, 0, \ldots, 0, 1)$. Notice that the revenue induced by $\mathbf{x}^\star$ is $U(\mathbf{x}^\star) = r(q(T-1, 1)) = 0.5$. Therefore,

$$\text{RPAR} = \frac{\max_{\mathbf{x}} U(\mathbf{x})}{U(\bar{\mathbf{x}})} > \frac{0.5}{Tr(q(\frac{T-1}{2}, 1))} = \frac{1 + e^{\xi\left(q(T-1)-q(\frac{T-1}{2})\right)}}{2T}$$
$$> \frac{e^{\xi\left(q(T-1)-q(\frac{T-1}{2})\right)}}{2T} = \frac{1 + e^{\ln(2TM-1)}}{2T} = M.$$

Notice that it holds that

$$L_r = \max_{p \in [0,1]} \frac{dr}{dp} = \max_{p \in [0,1]} r(p)(1 - r(p))\xi \le \frac{\xi}{4}.$$

For $T = 10$, we get that $L_r \approx 15.26 \ln(M)$. This completes the proof of Proposition 3.2. □

**Proof of Proposition 3.3.** Let $T \in \mathbb{R}_{>0}$ and consider the instance $a(\mathcal{D}) = \frac{\mathcal{D}}{T^3}$, $\gamma = 1$, $\beta = 1$, $w^s = \frac{1}{T}$ and $r(p) = p$.

Notice that the utility of the users from GenAI is bounded by

$$w_t^g(\mathbf{x}) = a(\mathcal{D}_t)x_t = \frac{\mathcal{D}}{T^3}x_t < \frac{T}{T^3} = \frac{1}{T^2} \le \frac{1}{T} = w^s.$$

Furthermore, we can bound the proportions by

$$p_t(\bar{\mathbf{x}}) = \frac{1}{1 + e^{\beta(w^s - w_t^g)}} > \frac{1}{1 + e^{\beta w^s}}.$$

Therefore, the users' social welfare satisfies that

$$w_t(\bar{\mathbf{x}}) = w_t^g(\bar{\mathbf{x}})p_t(\bar{\mathbf{x}}) + (1 - p_t(\bar{\mathbf{x}}))w^s$$
$$\le \frac{1}{T^2}p_t(\bar{\mathbf{x}}) + (1 - p_t(\bar{\mathbf{x}}))w^s$$
$$\le \frac{1}{T^2}\frac{1}{1 + e^{\beta w^s}} + (1 - \frac{1}{1 + e^{\beta w^s}})w^s$$
$$= \frac{1}{T^2}\frac{1}{1 + e^{\frac{\beta}{T}}} + (1 - \frac{1}{1 + e^{\frac{\beta}{T}}})w^s.$$

Next, denote $\tilde{\mathbf{x}} = (0, 0, \ldots 0)$, the strategy for which GenAI does not answer any query. Therefore, by definition it holds that $w_t(\tilde{\mathbf{x}}) = w^s$.

We now bound the price of anarchy:

$$\text{WPAR} = \frac{\max_{\mathbf{x}} W(\mathbf{x})}{W(\bar{\mathbf{x}})} \geq \frac{W(\tilde{\mathbf{x}})}{W(\bar{\mathbf{x}})}$$

$$= \frac{Tw^s}{W(\bar{\mathbf{x}})} \geq \frac{T}{T} \frac{w^s}{\frac{1}{T^2}\frac{1}{1+e^{\frac{\beta}{T}}} + (1 - \frac{1}{1+e^{\frac{\beta}{T}}})w^s}$$

$$= \frac{1}{\frac{\beta}{T}\frac{1}{1+e^{\frac{\beta}{T}}} + (1 - \frac{1}{1+e^{\frac{\beta}{T}}})}.$$

Notice that $\frac{1}{2-\varepsilon} > 0.5$. Next, denote $h(T) = \frac{\beta}{T}\frac{1}{1+e^{\frac{\beta}{T}}} + (1 - \frac{1}{1+e^{\frac{\beta}{T}}})$. Observe that $h(t)$ is continuous in $T$ and satisfies the following properties:

1. $h(1) = 1$,

2. $\lim_{T \to \infty} h(T) \to 0.5$.

Therefore, by the intermediate value theorem, there exists $T_0$ such that $h(T_0) = \frac{1}{2-\varepsilon}$. Furthermore,

$$\frac{dh}{dT} = -\frac{\beta}{T^2}\frac{1}{1+e^{\frac{\beta}{T}}} - \frac{\beta}{T}\frac{1}{1+e^{\frac{\beta}{T}}}\left(1 - \frac{1}{1+e^{\frac{\beta}{T}}}\right)\frac{\beta}{T^2} + \frac{1}{1+e^{\frac{\beta}{T}}}\left(1 - \frac{1}{1+e^{\frac{\beta}{T}}}\right)\frac{\beta}{T^2}$$

$$= -\frac{\beta}{T}\frac{1}{1+e^{\frac{\beta}{T}}}\left(1 - \frac{1}{1+e^{\frac{\beta}{T}}}\right)\frac{\beta}{T^2} - \frac{1}{1+e^{\frac{\beta}{T}}}\frac{1}{1+e^{\frac{\beta}{T}}}\frac{\beta}{T^2} < 0;$$

hence, for every $T > T_0$ it holds that

$$\text{WPAR} \geq \frac{1}{h(T)} \geq \frac{1}{h(T_0)} = \frac{1}{\frac{1}{2-\varepsilon}} = 2 - \varepsilon.$$

This completes the proof of Proposition 3.3. $\qquad\square$

**Theorem B.1.** *For every $M \in \mathbb{R}_{\geq 0}$ there exists an instance I with $PoA(I) > M$.*

**Proof of Theorem B.1.** Let $T \in \mathbb{R}_{>0}$ and Consider the instance $a(\mathcal{D}) = \frac{\mathcal{D}}{T}, \gamma = 1, \beta = 3, w^s = \frac{1}{T}$. We let $r(p)$ be the step function

$$r(p) = \begin{cases} 1 & p \geq q(T-1, 1) \\ 0 & \text{Otherwise} \end{cases}.$$

The purpose of choosing $r(p)$ as a step function is to show that GenAI's revenue-maximizing strategy is $(0, \ldots, 0, 1)$. Notice that we can also represent this function as a sigmoid $r(p) \approx \frac{1}{1+e^{\xi(q(T-1,1)-p)}}$ for $\xi \to \infty$.

Notice that in each turn, the maximal amount of data that can be generated is 1, which occurs for $x_t = 0$. Therefore, for $T - 1$ rounds, the maximum amount of data that can be generated is $T - 1$, which is induced by the strategy that uses $x_t = 0$ for every $t \leq T - 1$. Answering any query before round $T$ results in $r(p_t) = 0$ for every $t \in [T]$. Therefore, GenAI's optimal strategy is:

$$x_t^\star = \begin{cases} 0 & t < T \\ 1 & \text{Otherwise} \end{cases}.$$

We now evaluate the welfare for the schemes $\mathbf{x}^\star$ and $\bar{\mathbf{x}}$. We start with $\mathbf{x}^\star$:

$$W(\mathbf{x}^\star) = \sum_{t=1}^{T-1} w_t(\mathbf{x}^\star) + w_T(\mathbf{x}^\star) = (T-1)w_1(\mathbf{x}^\star) + w_T(\mathbf{x}^\star) \leq (T-1)w^s + 1 \leq Tw^s + 1 = 2.$$

We move on to evaluate the social welfare induced by $\bar{\mathbf{x}}$. First, notice that for every $T \geq 1$ it holds that

$$p_t(\bar{\mathbf{x}}) \geq p_1(\bar{\mathbf{x}}) = q(0,1) = \frac{1}{1 + e^{\beta(w^s - a(0))}} = \frac{1}{1 + e^{\frac{\beta}{T}}} \geq \frac{1}{1 + e^{\beta}} > 0.04.$$

Similarly, we develop an upper bound on the proportions:

$$p_t(\bar{\mathbf{x}}) = \frac{1}{1 + e^{\beta(w^s - a(\mathcal{D}_t(\bar{\mathbf{x}})))}} = \frac{1}{1 + e^{\beta\left(\frac{1}{T} - \frac{\mathcal{D}_t(\bar{\mathbf{x}})}{T}\right)}} < \frac{1}{1 + e^{-\beta\frac{T}{T}}} < 0.96.$$

Using the bound on the proportions, we can get a lower bound on the total amount of data at each round

$$\mathcal{D}_t(\bar{\mathbf{x}}) = \sum_{t'=1}^{t-1}(1 - p_{t'}(\bar{\mathbf{x}})) > 0.04(t-1).$$

This allows us to evaluate the minimal welfare induced by strategy $\bar{\mathbf{x}}$:

$$W(\bar{\mathbf{x}}) = \sum_{t=1}^{T} p_t(\bar{\mathbf{x}})w_t^g(\bar{\mathbf{x}}) + (1 - p_t(\bar{\mathbf{x}}))w^s > \sum_{t=1}^{T} p_t(\bar{\mathbf{x}})w_t^g(\bar{\mathbf{x}}) > 0.04\sum_{t=1}^{T} w_t^g(\bar{\mathbf{x}})$$

$$= 0.04\sum_{t=1}^{T} a(\mathcal{D}_t(\bar{\mathbf{x}})) > 0.04\sum_{t=1}^{T} \frac{0.04(t-1)}{T} = \frac{0.04^2}{2}(T-1).$$

We are now ready to plug everything we calculated so far into the definition of the PoA.

$$PoA = \frac{\max_{\mathbf{x}} W(\mathbf{x})}{\min_{\mathbf{x} \in \max U(\mathbf{x})} W(\mathbf{x})} = \frac{\max_{\mathbf{x}} W(\mathbf{x})}{W(\mathbf{x}^\star)}$$

$$\geq \frac{W(\bar{\mathbf{x}})}{W(\mathbf{x}^\star)} > \frac{\frac{0.04^2}{2}(T-1)}{2} = \frac{0.04^2}{4}(T-1).$$

Therefore, for every $T > \frac{4M}{0.04^2} + 1$, it holds that

$$PoA > \frac{0.04^2}{4}\left(\frac{4M}{0.04^2} + 1 - 1\right) = M.$$

This completes the proof of Theorem B.1. $\qquad\square$

## C. Proofs Omitted from Section 4

### C.1. Proofs Omitted from Subsection 4.1

**Proof of Proposition 4.2.** We take the derivative of $f(\mathcal{D}, x)$:

$$\frac{df(\mathcal{D}, x)}{d\mathcal{D}} = 1 - x\frac{dq(\mathcal{D}, x)}{d\mathcal{D}}$$

$$= 1 - \beta x^2 q(\mathcal{D}, x)(1 - q(\mathcal{D}, x))\frac{da(\mathcal{D})}{d\mathcal{D}}.$$

Notice that $q(d, x) \in [0, 1]$ for every $d \in \mathbb{R}_{\geq 0}$ and $x \in [0, 1]$. Furthermore, the expression $q(1 - q)$ has one maximum point at $q = 0.5$, therefore

$$\frac{df(\mathcal{D}, x)}{d\mathcal{D}} = 1 - \beta x^2 q(\mathcal{D}, x)\left(1 - q(\mathcal{D}, x)\right)\frac{da(\mathcal{D})}{d\mathcal{D}}$$

$$\geq 1 - \frac{\beta x^2}{4}\frac{da(\mathcal{D})}{d\mathcal{D}} \geq 1 - \frac{\beta}{4}\frac{da(\mathcal{D})}{d\mathcal{D}} > 0.$$

This completes the proof of Proposition 4.2. □

**Proof of Theorem 4.1.** We first show that if $y < x_\tau$ then $\mathcal{D}_{\tau+1}(\mathbf{x}^\tau) > \mathcal{D}_{\tau+1}(\mathbf{x})$. By definition of $\mathbf{x}^\tau$ and $\mathbf{x}$ it holds that $\mathcal{D}_t(\mathbf{x}^\tau) = \mathcal{D}_t(\mathbf{x})$ for every $t \leq \tau$. Next, notice that if $y < x_\tau$ then

$$p_\tau(\mathbf{x}^\tau) = x_\tau^\tau \frac{e^{\beta a(d_\tau(\mathbf{x}^\tau))x_\tau^\tau}}{e^{\beta a(d_\tau(\mathbf{x}^\tau))x_\tau^\tau} + e^{\beta w^s}}$$

$$= y\frac{e^{\beta a(d_\tau(\mathbf{x}^\tau))y}}{e^{\beta a(d_\tau(\mathbf{x}^\tau))y} + e^{\beta w^s}} < x_\tau \frac{e^{\beta a(d_\tau(\mathbf{x}))x_\tau}}{e^{\beta a(d_\tau(\mathbf{x}))x_\tau} + e^{\beta w^s}} = p_\tau(\mathbf{x});$$

therefore, it holds that

$$\mathcal{D}_{\tau+1}(\mathbf{x}^\tau) = \mathcal{D}_\tau(\mathbf{x}^\tau) + (1 - p_\tau(\mathbf{x}^\tau))$$

$$> \mathcal{D}_\tau(\mathbf{x}) + (1 - p_\tau(\mathbf{x})) = \mathcal{D}_{\tau+1}(\mathbf{x}).$$

Next, we use the following proposition to show that $\mathcal{D}_{\tau+1}(\mathbf{x}^\tau) > \mathcal{D}_{\tau+1}(\mathbf{x})$.

**Proposition C.1.** *Let $\tau \in [T]$ and $\mathbf{x}$, $\tilde{\mathbf{x}}$ be two selective response strategies such that $x_t = \tilde{x}_t$ for every $t \geq \tau$. If $\mathcal{D}_t(\mathbf{x}) > \mathcal{D}_t(\tilde{\mathbf{x}})$ and $\frac{da(\mathcal{D})}{d\mathcal{D}} < \frac{4}{\beta}$ Then for every $t \geq \tau$ it holds that $\mathcal{D}_t(\mathbf{x})_t > \mathcal{D}_t(\tilde{\mathbf{x}})_t$ and $p_t(\mathbf{x}) \geq p_t(\tilde{\mathbf{x}})$ where inequality holds only if $x_t = 0$.*

Thus, by Proposition C.1 it holds that $p_t(\mathbf{x}^\tau) \geq p_t(\mathbf{x})$. This completes the proof of Theorem 4.1. □

**Proof of Proposition C.1.** We prove our claim by proving a slightly stronger version using induction over the rounds. In addition to the original claim, we also prove that $\mathcal{D}_t(\mathbf{x}) > \mathcal{D}_t(\tilde{\mathbf{x}})$ for every $t \geq \tau$. We start with the base case $t = \tau$. Notice that $p_\tau(\mathbf{x}) = q(\mathcal{D}_\tau(\mathbf{x}), x_\tau)$ and $p_\tau(\tilde{\mathbf{x}}) = q(\mathcal{D}_\tau(\tilde{\mathbf{x}}), \tilde{x}_\tau)$.

We now use the following lemma:

**Lemma C.2.** *For every $x \in [0, 1]$ and $\mathcal{D} \in \mathbb{R}_{\geq 0}$, it holds that $q(\mathcal{D}, x)$ satisfies $\frac{dq(\mathcal{D}, x)}{d\mathcal{D}} \geq 0$.*

Since $\mathcal{D}_\tau(\mathbf{x}) > \mathcal{D}_\tau(\tilde{\mathbf{x}})$ then from Lemma C.2 it holds that $p_\tau(\mathbf{x}) \geq p_\tau(\tilde{\mathbf{x}})$. Next, we show that $\mathcal{D}_{\tau+1}(\mathbf{x}) > \mathcal{D}_{\tau+1}(\tilde{\mathbf{x}})$. Notice that $\mathcal{D}_{\tau+1}(\tilde{\mathbf{x}}) = \mathcal{D}_\tau(\tilde{\mathbf{x}}) + (1 - p_\tau(\tilde{\mathbf{x}})) = f(\mathcal{D}_\tau(\tilde{\mathbf{x}}), \tilde{x}_\tau)$, similarly $\mathcal{D}_{\tau+1}(\mathbf{x}) = f(\mathcal{D}_\tau(\mathbf{x}), x_\tau)$. By Proposition 4.2 it holds that $f(\mathcal{D}, x_\tau)$ is monotonic increasing in $\mathcal{D}$. Therefore, $\mathcal{D}_\tau(\mathbf{x}) > \mathcal{D}_\tau(\tilde{\mathbf{x}})$ leads to $f(\mathcal{D}_\tau(\mathbf{x}), x_\tau) > f(\mathcal{D}_\tau(\tilde{\mathbf{x}}), \tilde{x}_\tau)$ and thus $\mathcal{D}_{\tau+1}(\mathbf{x}) > \mathcal{D}_{\tau+1}(\tilde{\mathbf{x}})$.

Assume the claim holds for $t - 1 > \tau$, and we prove it holds for $t$. Since it holds for $t - 1$, then $\mathcal{D}_t(\mathbf{x}) > \mathcal{D}_t(\tilde{\mathbf{x}})$. Therefore, by Lemma C.2 it holds that $p_t(\mathbf{x}) > p_t(\tilde{\mathbf{x}})$. Lastly, by Proposition 4.2 it holds that

$$\mathcal{D}_{t+1}(\mathbf{x}) = f(\mathcal{D}_{t+1}(\mathbf{x}), x_t) > f(\mathcal{D}_{t+1}(\tilde{\mathbf{x}}), \tilde{x}_t) = \mathcal{D}_{t+1}(\tilde{\mathbf{x}}).$$

This completes the proof of Proposition C.1. □

**Proof of Lemma C.2.** We take the derivative of $q(\mathcal{D}, x)$:

$$\frac{dq(\mathcal{D}, x)}{d\mathcal{D}} = \beta x^2 q(\mathcal{D}, x)(1 - q(\mathcal{D}, x))\frac{da(\mathcal{D})}{d\mathcal{D}}.$$

As we assume in the model, $\frac{da(\mathcal{D})}{d\mathcal{D}} \geq 0$. Furthermore, $q(\mathcal{D}, x) \in [0, 1]$ for every $x \in [0, 1]$, and therefore $\frac{dq(\mathcal{D}, x)}{d\mathcal{D}} \geq 0$. This completes the proof of Lemma C.2. □

---

**Algorithm 1** Approximately optimal Selective Response (ASR)

---

**Input:** $T, A, \varepsilon$

**Output:** $\mathbf{x}$

1:  $V(t, d) \leftarrow 0, \pi(t, d) \leftarrow 0$ for every $t \in [T + 1]$ and $d \in \{0, \varepsilon, \dots, T\}$
2:  **for** $t = T \dots 1$ **do**
3:      **for** $d \in \{0, \varepsilon, \dots, t - 1\}$ **do**
4:          $U(y) \leftarrow 0$ for every $y \in A$
5:          **for** $y \in A$ **do**
6:              $p \leftarrow y \frac{e^{\beta a(d)y}}{e^{\beta a(d)y} + e^{\beta w^s}}$
7:              $d' \leftarrow \lfloor d + (1 - p) \rfloor_\varepsilon$
8:              $v_d(y) \leftarrow r(p) + \gamma V(t + 1, d')$
9:          **end for**
10:         $V(t, d) \leftarrow \max_y v_d(y)$
11:         $\pi(t, d) \leftarrow \arg\max_y v_d(y)$
12:     **end for**
13: **end for**
14: extract $\mathbf{x}$ from $\pi$ starting at $t = 1, d = 0$
15: Return $\mathbf{x}$

---

## C.2. Proofs Omitted from Subsection 4.2

**Proof of Observation 4.3.** Consider the instance $a(\mathcal{D}) = 0.7(1 - e^{0.4\mathcal{D}}) + 0.3$, $\gamma = 1$, $r(p) = p$, $\beta = 41$ and $w^s = 0.66$. Let $T = 3$ and observe the revenue for the following schemes:

1. $\bar{\mathbf{x}} = (1, 1, 1)$.

2. $\mathbf{x}^1 = (0, 1, 1)$.

3. $\mathbf{x}^2 = (0, 0, 1)$.

4. $\mathbf{x}^3 = (1, 0, 1)$.

5. $\mathbf{x} = (0.04, 0.97, 1)$.

Notice that we do not consider schemes where $x_3 = 0$ since for any such scheme, the scheme which is identical at round $t = 1$, $t = 2$ and plays $x_3 = 1$ induces higher revenue; therefore, the revenue difference between $\mathbf{x}$ and the other schemes is as follows:

- $U(\mathbf{x}) - U(\bar{\mathbf{x}}) > 9.71 \cdot 10^{-6}$.

- $U(\mathbf{x}) - U(\mathbf{x}^1) > 9.71 \cdot 10^{-6}$.

- $U(\mathbf{x}) - U(\mathbf{x}^2) > 7.9 \cdot 10^{-6}$.

- $U(\mathbf{x}) - U(\mathbf{x}^3) > 7.89 \cdot 10^{-6}$.

This completes the proof of Observation 4.3. $\qquad\square$

**Proof of Theorem 4.4.** We denote $\Delta_t = (t - 1)\varepsilon$ and $U_t(\mathbf{x})$ the accumulated revenue from round $t$ until $T$ following scheme $\mathbf{x}$, formally $U_t(\mathbf{x}) = \sum_{i=t}^{T} \gamma^{i-t} p_i(\mathbf{x})$. We use the following lemma to show the relationship between $V(t, \lfloor d \rfloor_\varepsilon)$ and $U_t(\mathbf{x}^\star)$.

**Lemma C.3.** *Fix round $t \in [T]$. For every $d \in \{0, \varepsilon, \dots, T\}$ such that $|d - \mathcal{D}_t(\mathbf{x}^\star)| \le \Delta_t$ it holds that*

$$V(t, d) > U_t(\mathbf{x}^\star) - L_r \sum_{i=t}^{T} \Delta_i \gamma^{i-t}.$$

Notice that $\mathcal{D}_1 = 0$ by definition, and thus $U_1(\mathbf{x}^\star) = U(\mathbf{x}^\star)$. Therefore, Lemma C.3 suggests that

$$V(1,0) > U(\mathbf{x}^\star) - L_r \sum_{i=1}^{T} \Delta_i \gamma^{i-1}.$$

We use the following lemma to evaluate the differences between $U(\mathbf{x})$ and $V(1,0)$.

**Lemma C.4.** *Let* $(d_t)_{t=1}^{T}$ *be the sequence defined by* $d_t = 0$ *and* $d_{t+1} = \lfloor d_t + (1 - x_t q(d_t, x_t)) \rfloor_\varepsilon$. *Then, for every* $t \in [T]$ *it holds that* $d_t < \mathcal{D}_t(\mathbf{x})$.

Therefore, by Lemma C.2 it holds that

$$V(1,0) = \sum_{t=1}^{T} r(x_t q(d_t, x_t)) \leq \sum_{t=1}^{T} r(x_t q(\mathcal{D}_t(\mathbf{x}), x_t)) = \sum_{t=1}^{T} r(p_t(\mathbf{x})) = U(\mathbf{x}).$$

Thus, we can write:

$$U(\mathbf{x}) \geq V(1,0) > U(\mathbf{x}^\star) - L_r \sum_{i=1}^{T} \Delta_i \gamma^{i-1}.$$

To complete the proof of Theorem 4.4, we prove the following lemma.

**Lemma C.5.** *It holds that* $\sum_{i=1}^{T} \Delta_i \gamma^{i-1} < \varepsilon T^2$.

This completes the proof of Theorem 4.4. $\qquad\square$

**Proof of Lemma C.3.** We prove this lemma using backward induction, starting with the base case from round $T$. For that, we start by bounding the difference in proportions using the following lemma.

**Lemma C.6.** *Let* $d^1, d^2 \in \mathbb{R}_{\geq 0}$ *and* $y \in [0,1]$. *If* $d^1 < d^2$ *then* $0 \leq y(q(d^2, y) - q(d^1, y)) < d^2 - d^1$.

In round $T$ it holds that $|d - \mathcal{D}_T(\mathbf{x}^\star)| < \Delta_T$. Therefore, for every $y \in A$ it holds that

$$y\,|q(d,y) - q(\mathcal{D}_T(\mathbf{x}^\star), y)| < |d - \mathcal{D}_T(\mathbf{x}^\star)| < \Delta_T.$$

Let $d'(y) = \lfloor d + (1 - y q(d, y)) \rfloor_\varepsilon$. Consequently,

$$
\begin{aligned}
|V(T,d) - U_T(\mathbf{x}^\star)| &= \left| \max_{y \in A}\{r(yq(d,y)) + \gamma V(T+1, d'(y))\} - U_T(\mathbf{x}^\star) \right| \\
&= \left| \max_{y \in A} r(yq(d,y)) - U_T(\mathbf{x}^\star) \right| \\
&= \left| \max_{y \in A} r(yq(d,y)) - r(x_T^\star q(\mathcal{D}_T(\mathbf{x}^\star), x_T^\star)) \right| \\
&= \left| r(\max_{y \in A} yq(d,y)) - r(x_T^\star q(\mathcal{D}_T(\mathbf{x}^\star), x_T^\star)) \right| \\
&= L_r \left| \max_{y \in A} yq(d,y) - x_T^\star q(\mathcal{D}_T(\mathbf{x}^\star), x_T^\star) \right| \\
&< L_r \Delta_T.
\end{aligned}
$$

We finished with the base case and move on to the induction step. Assume the lemma is true for $t+1$ and we show it holds for round $t$.

according to the assumptions in the lemma, it holds that $|d - \mathcal{D}_t(\mathbf{x}^\star)| < \Delta_t$, therefore according to Lemma C.6, for every $y \in A$ it holds that

$$|r(yq(d, y)) - r(yq(\mathcal{D}_t(\mathbf{x}^\star), y))| \leq L_r |yq(d, y) - yq(\mathcal{D}_t(\mathbf{x}^\star), y)| < L_r |d - \mathcal{D}_t(\mathbf{x}^\star)| < L_r \Delta_t.$$

We use the next lemma to bound the difference in data at step $t + 1$.

**Lemma C.7.** *it holds that* $|f(\mathcal{D}_t(\mathbf{x}^\star), y) - \lfloor f(d, y) \rfloor_\varepsilon| < \Delta_{t+1}$.

Lemma C.7 suggests that the condition for the induction step holds, and therefore according to our induction step:

$$V(t + 1, \lfloor f(d, x_t^\star) \rfloor_\varepsilon) > U_{t+1}(\mathbf{x}^\star) - L_r \sum_{i=t+1}^{T} \Delta_i \gamma^{i-(t+1)};$$

therefore,

$$
\begin{aligned}
v_d(x_t^\star) &= x_t^\star q(d, x_t^\star) + \gamma V(t + 1, \lfloor f(d, x_t^\star) \rfloor_\varepsilon) \\
&> x_t^\star q(\mathcal{D}_t(\mathbf{x}^\star), x_t^\star) - L_r \Delta_t + \gamma V(t + 1, \lfloor f(d, x_t^\star) \rfloor_\varepsilon) \\
&> x_t^\star q(\mathcal{D}_t(\mathbf{x}^\star), x_t^\star) - L_r \Delta_t + \gamma \left( U_{t+1}(\mathbf{x}^\star) - L_r \sum_{i=t+1}^{T} \Delta_i \gamma^{i-(t+1)} \right) \\
&= x_t^\star q(\mathcal{D}_t(\mathbf{x}^\star), x_t^\star) + \gamma U_{t+1}(\mathbf{x}^\star) - L_r \Delta_t - \gamma L_r \sum_{i=t+1}^{T} \Delta_i \gamma^{i-(t+1)} \\
&= U_t(\mathbf{x}) - L_r \sum_{i=t}^{T} \Delta_i \gamma^{i-t}.
\end{aligned}
$$

Finally, it holds that

$$V(t, d) = \max_{y \in A} v_d(y) \geq v_d(x_t^\star) > U_t(\mathcal{D}, \mathbf{x}) - L_r \sum_{i=t}^{T} \Delta_i \gamma^{i-t}.$$

This completes the proof of Lemma C.3. $\qquad\square$

**Proof of Lemma C.6.** Since $d^2 > d^1$ then according to Proposition 4.2 it holds that

$$f(d^2, y) - f(d^1, y) > 0;$$

hence,

$$
\begin{aligned}
f(d^2, y) - f(d^1, y) &= d^2 + (1 - yq(d^2, y)) - d^1 - (1 - yq(d^1, y)) \\
&= d^2 - yq(d^2, y) - d^1 + yq(d^1, y) > 0.
\end{aligned}
$$

Rearranging the above inequality, we get that

$$y(q(d^2, a) - q(d^1, a)) < d^2 - d^1.$$

Furthermore, from Lemma C.2 it holds that $q(d^2, a) \geq q(d^1, a)$ and therefore we can summarize

$$0 \leq y(q(d^2, a) - q(d^1, a)) < d^2 - d^1.$$

This completes the proof of Lemma C.6. $\qquad\square$

**Proof of Lemma C.7.** We prove that for any $\mathcal{D} \in \mathbb{R}_{\geq 0}$ it holds that it holds that $|f(D, y) - \lfloor f(d, y) \rfloor_\varepsilon| < \Delta_{t+1}$.

First, we use the following lemma.

**Lemma C.8.** *Let $d^1, d^2 \in [0, T]$ then it holds that*

$$\left| f(d^1, y) - f(d^2, y) \right| < \left| d^1 - d^2 \right|.$$

Therefore, using Lemma C.8 we get

$$
\begin{aligned}
|f(\mathcal{D}, y) - \lfloor f(d, y) \rfloor_\varepsilon| &\leq |f(\mathcal{D}, y) - f(d, y)| + \varepsilon \\
&< |\mathcal{D} - d| + \varepsilon \\
&\leq \Delta_t + \varepsilon \\
&= (t - 1)\varepsilon + \varepsilon \\
&= t\varepsilon = \Delta_{t+1}.
\end{aligned}
$$

This completes the proof of Lemma C.7. $\qquad\square$

**Proof of Lemma C.8.** Assume without loss of generality that $d^1 < d^2$. Therefore, according to Proposition 4.2, for every $y \in A$ it holds that $f(d^1, y) < f(d^2, y)$. Furthermore, from Lemma C.2 it holds that $q(d^1, y) \leq q(d^2, a)$. Thus, we can write:

$$
\begin{aligned}
\left| f(d^2, y) - f(d^1, y) \right| &= f(d^2, y) - f(d^1, y) \\
&= d^2 + (1 - yq(d^2, y)) - d^1 - (1 - yq(d^1, y)) \\
&= d^2 - d^1 - yq(d^2, y) + yq(d^1, y) \\
&\leq d^2 - d^1 - yq(d^1, y) + yq(d^1, y) \\
&= d^2 - d^1 \\
&\leq \left| d^2 - d^1 \right|.
\end{aligned}
$$

This completes the proof of Lemma C.8. $\qquad\square$

**Proof of Lemma C.4.** By definition and by Proposition 4.2 it holds that:

$$d_{t+1} = \lfloor d_t + (1 - x_t q(d_t, x_t)) \rfloor_\varepsilon = \lfloor f(d_t, x_t) \rfloor_\varepsilon \leq f(d_t, x_t) \leq f(\mathcal{D}_t(\mathbf{x})) = \mathcal{D}_{t+1}(\mathbf{x}).$$

This completes the proof of Lemma C.4. $\qquad\square$

**Proof of Lemma C.5.** Since $\gamma \leq 1$ it holds that $\sum_{i=1}^{T} \Delta_i \gamma^{i-1} \leq \sum_{i=1}^{T} \Delta_i$. Notice that we now have a sum of an arithmetic series and therefore

$$\sum_{i=1}^{T} \Delta_i = \varepsilon \sum_{i=1}^{T} (i - 1) = \varepsilon \sum_{i=0}^{T-1} i = \varepsilon \frac{T(T-1)}{2} < \varepsilon T^2.$$

This completes the proof of Lemma C.5. $\qquad\square$

**Proof of Theorem 4.6.** Denote $\mathbf{x}^\star = \max_{\mathbf{x}'} U(\mathbf{x}')$ and we define the following $T$ different strategies $\{\mathbf{x}(i)\}_{i=1}^{T+1}$ such that

$$x(i)_t = \begin{cases} \lfloor x_t^\star \rfloor_\delta & t \geq i \\ x_t^\star & \text{Otherwise} \end{cases}.$$

Notice that by definition $\mathbf{x}(T+1) = \mathbf{x}^\star$. Furthermore, observe that the strategies $\mathbf{x}(i)$ and $\mathbf{x}(i+1)$ for every $i \in [T]$ differ only in round $i$. The following lemma bound the difference between strategy $\mathbf{x}(i)$ and strategy $\mathbf{x}(i+1)$.

**Lemma C.9.** *For every $i \in [T]$ it holds that*

$$|U(\mathbf{x}(i)) - \mathbf{x}(i+1)| \leq \frac{\gamma^{i-1}}{1-\gamma} \left( \frac{7}{4}\beta + 1 \right) L_r \delta.$$

Observe that

$$|U(\mathbf{x}(1)) - U(\mathbf{x}(T+1))| \leq \sum_{i=1}^{T} |U(\mathbf{x}(i)) - U(\mathbf{x}(i+1))|.$$

Therefore, by lemma C.9 we get that

$$|U(\mathbf{x}(1)) - U(\mathbf{x}(T+1))| \leq \sum_{i=1}^{T} \frac{\gamma^{i-1}}{1-\gamma} \left( \frac{7}{4}\beta + 1 \right) L_r \delta \leq \frac{7\beta + 1}{4(1-\gamma)^2} L_r \delta.$$

Lastly, notice that $U(\mathbf{x}^\star) \geq \max_{\mathbf{x}' \in A_\delta^T} U(\mathbf{x}') \geq U(\mathbf{x}(1))$. Therefore, we can write:

$$|U(\mathbf{x}^\star) - U(\mathbf{x})| = \left| U(\mathbf{x}^\star) - \max_{\mathbf{x}' \in A_\delta^T} U(\mathbf{x}') + \max_{\mathbf{x}' \in A_\delta^T} U(\mathbf{x}') - U(\mathbf{x}) \right|$$

$$\leq \left| U(\mathbf{x}^\star) - \max_{\mathbf{x}' \in A_\delta^T} U(\mathbf{x}') \right| + \left| \max_{\mathbf{x}' \in A_\delta^T} U(\mathbf{x}') - U(\mathbf{x}) \right|$$

$$\leq \left| U(\mathbf{x}^\star) - U(\mathbf{x}(1)) + U(\mathbf{x}(1)) - \max_{\mathbf{x}' \in A_\delta^T} U(\mathbf{x}') \right| + \left| \max_{\mathbf{x}' \in A_\delta^T} U(\mathbf{x}') - U(\mathbf{x}) \right|$$

$$\leq |U(\mathbf{x}^\star) - U(\mathbf{x}(1))| + \left| \max_{\mathbf{x}' \in A_\delta^T} U(\mathbf{x}') - U(\mathbf{x}) \right|$$

$$\leq \frac{7\beta + 1}{4(1-\gamma)^2} L_r \delta + \varepsilon L_r T^2.$$

This completes the proof of Theorem 4.6. $\qquad\qquad\square$

**Proof of Lemma C.9.** By definition, $x(i)_t = x(i+1)_t$ for every $t < i$ and therefore $\mathcal{D}_t(\mathbf{x}(i)) = \mathcal{D}_t(\mathbf{x}(i+1))$.

Next, we use the following lemma:

**Lemma C.10.** *For every $\mathcal{D}^1, \mathcal{D}^2 \in [0, T]$ and $x, x' \in [0, 1]$ it holds that*

$$|q(\mathcal{D}, x) - q(\mathcal{D}, x')| = q(\mathcal{D}, x) (1 - q(\mathcal{D}, x')) \left| 1 - e^{\beta(x' a(\mathcal{D}) - x a(\mathcal{D}))} \right|.$$

Notice that in our case, for every $|x(i)_i - x(i+1)_i| < \delta$. Therefore,

$$|q(\mathcal{D}_i(\mathbf{x}(i)), x(i)_i) - q(\mathcal{D}_i(\mathbf{x}(i+1)), x(i+1)_i)|$$
$$= q(\mathcal{D}_i(\mathbf{x}(i)), x(i)_i) (1 - q(\mathcal{D}_i(\mathbf{x}(i+1)), x(i+1)_i)) \left| 1 - e^{\beta a(\mathcal{D}_i(\mathbf{x}(i)))(x(i+1)_i - x(i)_i)} \right|$$
$$\leq \left| 1 - e^{\beta a(\mathcal{D}_i(\mathbf{x}(i)))(x(i+1)_i - x(i)_i)} \right|$$
$$\leq \frac{7}{4}\beta\delta.$$

Where the last inequality follows from $|1 - e^x| \leq \frac{7x}{4}$ for every $|x| < 1$. Next, notice that for every $\mathcal{D} \in \mathbb{R}_{\geq 0}$ and $x, x' \in [0, 1]$ such that $|x - x'| \leq \delta$ it holds that

$$
\begin{aligned}
|xq(\mathcal{D}, x) - x'q(\mathcal{D}, x')| &= |xq(\mathcal{D}, x) - (x' - x + x)\, q(\mathcal{D}, x')| \\
&= |xq(\mathcal{D}, x) - xq(\mathcal{D}, x') - (x' - x)\, q(\mathcal{D}, x')| \\
&\leq |xq(\mathcal{D}, x) - xq(\mathcal{D}, x')| + |x' - x|\, q(\mathcal{D}, x') \\
&\leq |xq(\mathcal{D}, x) - xq(\mathcal{D}, x')| + |x' - x| \\
&\leq \left(\frac{7}{4}\beta + 1\right)\delta.
\end{aligned}
$$

Therefore, by Corollary 6.2 it holds that

$$
\begin{aligned}
|U(\mathbf{x}(i)) - U(\mathbf{x}(i+1))| &\leq \gamma^{i-1}\, |r(p_i(\mathbf{x}(i))) - r(p_i(\mathbf{x}(i+1)))| + L_r\gamma^i \frac{|p_i(\mathbf{x}(i)) - p_i(\mathbf{x}(i+1))|}{1 - \gamma} \\
&\leq \gamma^{i-1} L_r\, |p_i(\mathbf{x}(i)) - p_i(\mathbf{x}(i+1))| + L_r\gamma^i \frac{|p_i(\mathbf{x}(i)) - p_i(\mathbf{x}(i+1))|}{1 - \gamma} \\
&= \gamma^{i-1} L_r\, |p_i(\mathbf{x}(i)) - p_i(\mathbf{x}(i+1))| \frac{1}{1 - \gamma} \\
&\leq \frac{\gamma^{i-1}}{1 - \gamma}\left(\frac{7}{4}\beta + 1\right) L_r\delta.
\end{aligned}
$$

This completes the proof of Lemma C.9. □

**Proof of Lemma C.10.** This lemma is a special case of Lemma E.7 and is hence omitted. □

### C.3. Proofs Omitted from Subsection 4.3

**Proof of Theorem 4.7.** The proof is constructed in 5 parts. First, we simplify and write our problem explicitly. Then, we define an approximation to our problem and build an MDP to describe it. The third step is to show that our approximation problem can be viewed as an instance of the problem in (Ben-Porat et al., 2024) and thus has an optimal solution. In the last step, we calculate the gap between the optimal solution of the approximated problem and the optimal solution of our original problem.

**Step 1.** We start by rewriting Problem P1. Notice that the welfare at each round can be written as

$$
w_t(\mathbf{x}) = p_t(\mathbf{x})a(\mathcal{D}_t(\mathbf{x})) + (1 - p_t(\mathbf{x}))w^s = p_t(\mathbf{x})\left(a(\mathcal{D}_t(\mathbf{x})) - w^s\right) + w^s.
$$

Therefore, the social welfare can be expressed as $W(\mathbf{x}) = Tw^s + \sum_{t=1}^{T} p_t(\mathbf{x})\left(a(\mathcal{D}_t(\mathbf{x})) - w^s\right)$. By denoting $W^1 = W - Tw^s$ we can rewrite our problem as

$$
\max_{\mathbf{x}} \sum_{t=1}^{T} r(p_t(\mathbf{x}))
$$
$$
s.t \quad \sum_{t=1}^{T} p_t(\mathbf{x})\left(a(\mathcal{D}_t(\mathbf{x})) - w^s\right) \geq W^1.
$$

**Step 2.** We now build a graph to represent an approximation of our problem. Notice that the maximum amount of data that can be generated in each round is 1 and therefore $\mathcal{D}_t(\mathbf{x}) < T$ for every $t \in [T]$ and scheme $\mathbf{x}$. Therefore, given $\varepsilon > 0$, we discretize all the available data values by increments of $\frac{T}{\varepsilon}$. We know describe the components of our graph. Our graph is a deterministic MDP with an underlying layered graph as follows: let $S = \{S_1, \ldots S_{T+1}\}$ the set of all states where $S_t = \{s_t^0, s_t^\varepsilon, \ldots, s_t^{\varepsilon T}\}$ denote the state in the $t$'th layer where $s_t^d$ represents the state where GenAI is in round $t$

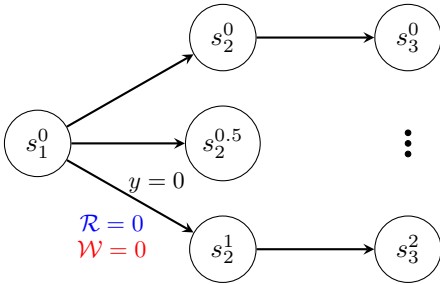

Figure 3: Example of a constructed graph with a discretization factor of $\varepsilon = 0.5$.

with $d$ data. The set of actions is $A$, and there are 2 reward functions defined for each state-action pair. First is defined by $\mathcal{R}(s_t^d, y) = \lfloor r(yq(d, y)) \rfloor_\varepsilon$ while the second is $\mathcal{W}(s_t^d, y) = \lfloor yq(d, y)(a(d) - w^s) \rfloor_\varepsilon$. Next, we let $\mathcal{T}(s, y, s')$ denote the transition function, which denotes the probability of reaching state $s'$ by playing $y$ in state $s$. The transition function in our MDP is deterministic and defined by

$$\mathcal{T}(s_t^d, y, s_{t'}^{d'}) = \begin{cases} 1 & t' = t + 1 \text{ and } d' = \lfloor d + 1 - yq(d, y) \rfloor_\varepsilon \\ 0 & \text{Otherwise} \end{cases}.$$

In terms of graphs, the states are analogous to vertices, and $T(s, y, s') = 1$ specifies an edge from state $s$ to state $s'$. An illustration of this graph for $\varepsilon = 0.5$ is presented in Figure 3.

By the construction of the layered graph, the horizon is $T + 1$, and GenAI starts at state $s_1^0$. We define policy $\pi : S \to A$ to be the mapping between each state and the action GenAI should take in that state. For a deterministic MDP, a policy is equivalent to a path $\tau$, which in our case is a sequence of $T$ edges starting from state $s_1^0$ and leading to a state in $S_{T+1}$. Notice that each edge represents a state and an action from that state, and therefore, path $\tau$ can also be defined as a sequence of state-action pairs.

The problem we aim to solve using the graph is the following problem:

$$\max_\tau \sum_{(s,y)\in\tau} \mathcal{R}(s, y) \tag{P3}$$

$$s.t \sum_{(s,y)\in\tau} \mathcal{W}(s, y) \geq W^1.$$

**Step 3.** Recall that a selective response strategy is a vector that specifies the portion of queries GenAI should answers. We denote $\pi^{\mathbf{x}}$ that follows scheme $\mathbf{x}$, that is $\pi^{\mathbf{x}}$ assigns the same action to all states at round $t$ as $x_t$, formally $\pi^{\mathbf{x}}(s_t^d) = x_t$ for every $t \in [T]$ and $d \in \{0, \varepsilon, \ldots T\}$.

We now introduce some notations that we use in this step. First, we denote $U_t(\mathbf{x})$ and $W_t(\mathbf{x})$ the accumulative revenue and welfare from round $t$ until $T$, following scheme $\mathbf{x}$. Formally

- $U_t(\mathbf{x}) = r(x_t q(\mathcal{D}_t(\mathbf{x}), x_t)) + U_{t+1}(\mathbf{x})$,

- $W_t(\mathbf{x}) = x_t q(\mathcal{D}_t(\mathbf{x}), x_t)(a(\mathcal{D}_t(\mathbf{x})) - w^s) + W_{t+1}(\mathbf{x})$.

We now define the analog of $U_t$ and $W_t$ in our MDP. Let $V^G(\pi, s)$ denote the sum of rewards with respect to reward function $\mathcal{R}$, following policy $\pi$ and starting at state $s$ in our MDP. Similarly, denote $V^W(\pi, s)$ the sum of rewards with respect to reward function $\mathcal{W}$. Formally

- $V^G(\pi, s_t^d) = \mathcal{R}(s_t^d, \pi(s)) + V^G(\pi, s_{t+1}^{d'})$,

- $V^W(\pi, s_t^d) = \mathcal{W}(s_t^d, \pi(s)) + V^W(\pi, s_{t+1}^{d'})$.

We are now ready to compare the values of the revenue and social welfare following a given selective response strategy to those from the MDP. Let $\mathbf{x}$ be an arbitrary selective response strategy and $M = \max\{1, L_r\}$. We use the following lemma.

**Lemma C.11.** *Fix round $t \in [T]$, then for every $d \in \{0, \varepsilon, \dots T\}$ such that $|d - \mathcal{D}_t(\mathbf{x})| < (t-1)\varepsilon$ it holds that*

- $\left| V^G(\pi^{\mathbf{x}}, s_t^d) - U_t(\mathbf{x}) \right| \leq \varepsilon M \sum_{i=t}^{T} i,$

- $\left| V^W(\pi^{\mathbf{x}}, s_t^d) - W_t(\mathbf{x}) \right| \leq \varepsilon(L_a + 1) \sum_{i=t}^{T} i.$

Notice that $\sum_{i=1}^{T} i < T^2$ and therefore we can simplify the summations in Lemma C.11.

Given the optimal selective response strategy $\mathbf{x}^\star$, Lemma C.11 suggests that

- $V^G(\pi^{\mathbf{x}^\star}, s_1^0) \geq U_t(\mathbf{x}^\star) - \varepsilon M T^2,$

- $V^W(\pi^{\mathbf{x}^\star}, s_1^0) \geq W_t(\mathbf{x}^\star) - (L_a + 1)\varepsilon T^2.$

We finished Step 2 and now move on to develop the machinery to find the selective response strategy that gives us the guarantees of our theorem.

**Step 4.** We define the Weight-Constrained Shortest Path (WCSSP) (Garey & Johnson, 1979). Given a weighted graph $G = (V, E)$ with weights $\{w_e\}_{e \in E}$, costs $\{c_e\}_{e \in E}$ and a maximum weight $W \in \mathbb{R}$, the problem is to find the path with the least cost while keeping the total weights below $W$. Let $\tau$ denote a path, and therefore, the WCSSP problem is defined as

$$\min_{\tau} \sum_{e \in \tau} c_e \tag{P4}$$
$$\sum_{e \in \tau} w_e \leq \tilde{W}.$$

Problem P3 can be seen as an instance of Problem (P4) for by setting:

- $c(s, y) = -\mathcal{R}(s, y),$

- $w(s, y) = -\mathcal{W}(s, y),$

- $\tilde{W} = -W^1.$

To account for the approximation error in the welfare due to calculating it using the MDP, we choose $\tilde{W} = -\left(W^1 - \varepsilon T^2(L+1)\right).$

Problem (P4) is a known NP-Hard problem with a reduction to the PARS-MDP problem (Ben-Porat et al., 2024) with a deterministic transition function. The PARSE-MDP problem is defined over an MDP with two reward functions $R^A, R^P : S \times A \to \mathbb{R}_{\geq 0}$ and a budget $B \in \mathbb{R}_{\geq 0}$. The goal is to construct a new reward function $R^B : S \times A \to \mathbb{R}_{\geq 0}$ such that the total rewards over the whole MDP is less than $B$, and the induced policy that maximizes $R^A + R^B$ also maximizes $R^P$ under the constraint. Formally, the PARS-MDP is defined as follows:

$$\max_{R^B} V(\pi, R^P)$$
$$\sum_{s \in S, y \in A} R^B(s, y) \leq B \tag{P5}$$
$$R^B(s, y) \geq 0 \text{ for every } s \in S, y \in A(s)$$
$$\pi \in \mathcal{A}(R^A + R^B)$$

where $V(\pi, R^P)$ is the total sum of rewards from $R^P$ following policy $\pi$. Therefore, we make the following definitions to represent Problem (P3) as an instance of Problem (P5): First, denote $\tau^A$ the path that maximizes $R^A$, i.e $\tau^A \in \arg\max_\tau \sum_{s,y\in\tau} R^A(s,y)$. Notice that $\tau^A$ can be computed using standard methods which run in polynomial time with respect to the problem's parameters. Thus, we refer to $\tau^A$ as a known parameter and define the parameters of PARS-MDP as follows:

- $R^P(s,y) = \mathcal{R}(s,y)$,

- $R^A(s,y) = \mathcal{W}(s,y)$,

- $B = \sum_{s,y\in\tau^A} R^A(s,y) - \left(W^1 - \varepsilon T^2(L_a + 1)\right)$.

Notice that $\mathcal{W}, \mathcal{R}$ are in increments of $\varepsilon$ by the construction of our MDP. Therefore, we can use Theorem (5) from (Ben-Porat et al., 2024) to show that the optimal path of Problem (P3) can be found in polynomial time with respect to the problem's parameters.

**Theorem C.12.** *There is a known algorithm to compute the path $\tilde{\tau}$ which induces*

- $\sum_{s,y\in\tilde{\tau}} R^P(s,y) = \max_\tau \sum_{s,y\in\tau} R^P(s,y)$,

- $\sum_{s,y\in\tilde{\tau}} R^A(s,y) \geq \sum_{s,y\in\tau^A} R^A(s,y) - B$.

*in time $O(\frac{|S||A|T}{\varepsilon} \log(\frac{|A|T}{\varepsilon}))$.*

Using the terms from our MDP, the solution from the algorithm in Theorem C.12 guarantees

- $\sum_{s,y\in\tilde{\tau}} \mathcal{R}(s,y) = \max_\tau \sum_{s,y\in\tau} \mathcal{R}(s,y)$,

- $\sum_{s,y\in\tilde{\tau}} \mathcal{W}(s,y) \geq W^1 - \varepsilon T^2(L_a + 1)$.

Let $\tau^\star$ be the path corresponding to $\mathbf{x}^\star$. Notice that $\tau^\star$ guarantees

$$\sum_{s,y\in\tau^\star} \mathcal{W}(s,y) \geq W(x^\star) - \varepsilon T^2(L_a + 1) \geq W^1 - \varepsilon T^2(L_a + 1).$$

The path $\tau^\star$ is a possible solution of the PARS-MDP and therefore, by Theorem C.12, the path $\tilde{\tau}$ guarantees

- $\sum_{s,y\in\tilde{\tau}} \mathcal{R}(s,y) \geq \sum_{s,y\in\tau^\star} \mathcal{R}(s,y) \geq U(\mathbf{x}^\star) - \varepsilon M T^2$,

- $\sum_{s,y\in\tilde{\tau}} \mathcal{W}(s,y) \geq W^1 - \varepsilon T^2(L_a + 1)$.

**Step 5.** Let $\tilde{\mathbf{x}}$ be the selective response strategy corresponding to path $\tilde{\tau}$ and we compare the revenue and welfare when playing $\tilde{\mathbf{x}}$.

We begin with the following lemma.

**Lemma C.13.** *Fix scheme $\mathbf{x}$ Let $(d_t)_{t=1}^T$ be the sequence defined by $d_t = 0$ and $d_{t+1} = f^\varepsilon(d_t, x_t)$ then for every $t \in [T]$ it holds that $d_t < \mathcal{D}_t(\mathbf{x})$.*

Let $(d_t)_{t=1}^T$ be the sequence defined by $d_t = 0$ and $d_{t+1} = f^\varepsilon(d_t, \tilde{x}_t)$, then by Lemma C.2 we get that

$$\begin{aligned}
V^G(\pi^{\tilde{\mathbf{x}}}, s_1^0) = \sum_{t=1}^T \lfloor r(yq(d_t, \tilde{x}_t)) \rfloor_\varepsilon &\leq \sum_{t=1}^T r(yq(d_t, \tilde{x}_t)) \\
&\leq \sum_{t=1}^T r(yq(\mathcal{D}_t(\tilde{\mathbf{x}}), \tilde{x}_t)) = \sum_{t=1}^T r(p_t(\tilde{\mathbf{x}})) = U(\tilde{\mathbf{x}}).
\end{aligned}$$

Therefore, it holds that

$$U(\tilde{\mathbf{x}}) > V^G(\pi^{\tilde{\mathbf{x}}}, s_1^0) > U(\mathbf{x}^\star) - \varepsilon MT^2.$$

We move on to evaluate the welfare. Notice that the welfare is not monotonic in $p_t$. Instead of using our previous technique, we use Lemma C.11 and get that

$$\begin{aligned}
W(\tilde{\mathbf{x}}) &\geq V^W(\tilde{\mathbf{x}}, s_1^0) - \varepsilon T^2(L_a + 1) \\
&\geq W^1 - \varepsilon T^2(L_a + 1) - \varepsilon T^2(L_a + 1) \\
&= W^1 - 2\varepsilon T^2(L_a + 1).
\end{aligned}$$

This completes the proof of Theorem 4.7. $\qquad\square$

**Proof of Lemma C.11.** We begin by showing that there cannot be a large gap between the data accumulated in the original problem and the data according to our MDP following the same scheme. Let $\Delta_t = (t-1)\varepsilon$ and let $f^\varepsilon(d, y) = \lfloor f(d, y) \rfloor_\varepsilon = \lfloor d + 1 - yq(d, y) \rfloor_\varepsilon$ the data in the next round given that in the current round, GenAI started with $d$ data and played $y$. We use the following lemma to show that the accumulated data in the MDP cannot be too far from the accumulated in our original problem.

**Lemma C.14.** *Let $d \in \{0, \varepsilon, \ldots T\}$ and fix round $t \in [T]$. If $|d - \mathcal{D}_t(\mathbf{x})| < \Delta_t$ then $|f^\varepsilon(d, x_t) - \mathcal{D}_{t+1}(\mathbf{x})| < \Delta_{t+1}$.*

We now use backward induction to prove our lemma, starting at round $T$. Let $d \in \{0, \varepsilon, \ldots, T\}$ such that $|d - \mathcal{D}_T(\mathbf{x})| < \Delta_T$ and we use the following lemma:

**Lemma C.15.** *Let $d \in \{0, \varepsilon, \ldots T\}$ and fix round $t \in [T]$. If $|d - \mathcal{D}_t(\mathbf{x})| < \Delta_t$ then it holds that*

$$|\lfloor r(x_t q(d, x_t)) \rfloor_\varepsilon - r(x_t q(\mathcal{D}_t(\mathbf{x}), x_t))| \leq M\Delta_{t+1}.$$

Therefore, by Lemma C.15 we get that

$$\left| V^G(\pi^{\mathbf{x}}, s_T^d) - U_T(\mathbf{x}) \right| = |\lfloor r(x_T q(d, x_T)) \rfloor_\varepsilon - r(x_T q(\mathcal{D}_T(\mathbf{x}), x_T))| \leq M\Delta_{T+1}.$$

Similarly for $V^W$ and $W_T$. We use the following lemma.

**Lemma C.16.** *Let $d^1, d^2 \in \mathbb{R}_{\geq 0}$ and any $y \in [0, 1]$ then it holds that*

$$\left| yq(d^1, y)\left(a(d^1) - w^s\right) - yq(d^2, y)\left(a(d^2) - w^s\right) \right| \leq \left| d^1 - d^2 \right|(L_a + 1).$$

Therefore, it holds that:

$$\begin{aligned}
&\left| V^W(\pi^{\mathbf{x}}, s_T^d) - W_T(\mathbf{x}) \right| \\
&= |\lfloor x_T q(d, x_T)\left(a(d) - w^s\right) \rfloor_\varepsilon - x_T q(\mathcal{D}_T(\mathbf{x}), x_T)\left(a(\mathcal{D}_T(\mathbf{x})) - w^s\right)| \\
&\leq |x_T q(d, x_T)\left(a(d) - w^s\right) - x_T q(\mathcal{D}_T(\mathbf{x}), x_T)\left(a(\mathcal{D}_T(\mathbf{x})) - w^s\right)| + \varepsilon \\
&\leq |d - \mathcal{D}_T(\mathbf{x})|(L + 1) + \varepsilon \\
&\leq \Delta_T(L_a + 1) + \varepsilon \\
&< \Delta_{T+1}(L_a + 1).
\end{aligned}$$

We are done with the base case and can continue towards the induction step. Assume the lemma holds for round $t + 1$, and we prove it for round $t$.

We start with the revenue at round $t$. Let $d \in \{0, \varepsilon, \dots, T\}$ and denote $d' = f^{\varepsilon}(d, x_t)$. Then, for every $d$ such that $|d - \mathcal{D}_t(\mathbf{x})| < \Delta_t$ it holds that

$$\left| V^G(\pi^{\mathbf{x}}, s_t^d) - U_t(\mathbf{x}) \right| = \left| \lfloor r(x_t q(d, x_t)) \rfloor_{\varepsilon} + V^G(\pi^{\mathbf{x}}, s_{t+1}^{d'}) - r(x_t q(\mathcal{D}_t(\mathbf{x}), x_t)) - U_{t+1}(\mathbf{x}) \right|$$

$$\leq \left| \lfloor r(x_t q(d, x_t)) \rfloor_{\varepsilon} - r(x_t q(\mathcal{D}_t(\mathbf{x}), x_t)) \right| + \left| V^G(\pi^{\mathbf{x}}, s_{t+1}^{d'}) - U_{t+1}(\mathbf{x}) \right|.$$

We use Lemma C.15 to bound the first expression. Furthermore, notice that according to Lemma C.14 it holds that $V^G(\pi^{\mathbf{x}}, s_{t+1}^{d'})$ satisfies the conditions of our induction step. Therefore,

$$\left| V^G(\pi^{\mathbf{x}}, s_t^d) - U_t(\mathbf{x}) \right| \leq M\Delta_{t+1} + \sum_{i=t+2}^{T+1} M\Delta_i = M \sum_{i=t}^{T} \Delta_{i+1} = \varepsilon M \sum_{i=t}^{T} i.$$

We perform a similar calculation for the welfare:

$$\left| V^W(\pi^{\mathbf{x}}, s_t^d) - W_t(\mathbf{x}) \right|$$

$$= \left| \lfloor x_t q(d, x_t) \left( a(d) - w^s \right) \rfloor_{\varepsilon} + V^W(\pi^{\mathbf{x}}, s_{t+1}^{d'}) - x_t q(\mathcal{D}_t(\mathbf{x}), x_t) \left( a(\mathcal{D}_t(\mathbf{x})) - w^s \right) - W_{t+1}(\mathbf{x}) \right|$$

$$\leq \left| \lfloor x_t q(d, x_t) \left( a(d) - w^s \right) \rfloor_{\varepsilon} - x_t q(\mathcal{D}_t(\mathbf{x}), x_t) \left( a(\mathcal{D}_t(\mathbf{x})) - w^s \right) \right| + \left| V^W(\pi^{\mathbf{x}}, s_{t+1}^{d'}) - W_{t+1}(\mathbf{x}) \right|$$

$$\leq \Delta_{t+1}(L_a + 1) + (L_a + 1) \sum_{i=t+2}^{T+1} \Delta_i$$

$$= (L_a + 1) \sum_{i=t}^{T} \Delta_{i+1} = (L_a + 1)\varepsilon \sum_{i=t}^{T} i.$$

This completes the proof of Lemma C.11. □

**Proof of Lemma C.14.** This is a special case of Lemma C.7 and is hence omitted. □

**Proof of Lemma C.15.** By Lemma C.6, it holds that

$$\left| \lfloor r(x_t q(d, x_t)) \rfloor_{\varepsilon}) - r(x_t q(\mathcal{D}_T(\mathbf{x}), x_t)) \right| \leq \left| r(x_t q(d, x_t)) - r(x_t q(\mathcal{D}_T(\mathbf{x}), x_t)) \right| + \varepsilon$$

$$\leq L_r \left| x_t q(d, x_t) - x_t q(\mathcal{D}_T(\mathbf{x}), x_t) \right| + \varepsilon$$

$$\leq L_r \left| d - \mathcal{D}_T(\mathbf{x}) \right| + \varepsilon$$

$$\leq L_r \Delta_t + \varepsilon$$

$$\leq \Delta_{T+1} \max\{L_r, 1\}.$$

This completes the proof of Lemma C.15. □

**Proof of Lemma C.16.** We prove this by starting with the definition.

$$\left| y q(d^1, y) \left( a(d^1) - w^s \right) - y q(d^2, y) \left( a(d^2) - w^s \right) \right|$$

$$= \left| y \left( q(d^1, y) - q(d^2, y) + q(d^2, y) \right) \left( a(d^1) - w^s \right) - y q(d^2, y) \left( a(d^2) - w^s \right) \right|$$

$$\leq \left| y q(d^2, y) \left( a(d^1) - w^s \right) - y q(d^2, y) \left( a(d^2) - w^s \right) \right|$$

$$+ \left| y \left( q(d^1, y) - q(d^2, y) \right) \left( a(d^1) - w^s \right) \right|$$

$$\leq \left| y q(d^2, y) \left( a(d^1) - w^s - a(d^2) + w^s \right) \right| + \left| y \left( q(d^1, y) - q(d^2, y) \right) \left( a(d^1) - w^s \right) \right|$$

$$= \left| y q(d^2, y) \left( a(d^1) - a(d^2) \right) \right| + \left| y \left( q(d^1, y) - q(d^2, y) \right) \left( a(d^1) - w^s \right) \right|.$$

We use Lemma C.6 and therefore

$$\left| yq(d^1, y) \left( a(d^1) - w^s \right) - yq(d^2, y) \left( a(d^2) - w^s \right) \right|$$
$$\leq \left| yq(d^2, y) \left( a(d^1) - a(d^2) \right) \right| + \left| d^1 - d^2 \right| \left| a(d^1) - w^s \right|$$
$$= yq(d^2, y) \left| a(d^1) - a(d^2) \right| + \left| d^1 - d^2 \right| \left| a(d^1) - w^s \right|$$
$$\leq yq(d^2, y) L_a \left| d^1 - d^2 \right| + \left| d^1 - d^2 \right| \left| a(d^1) - w^s \right|.$$

Notice that $y, q(d^2, y), a(d^1), w^s \leq 1$ and thus we get that

$$\left| yq(d^1, y) \left( a(d^1) - w^s \right) - yq(d^2, y) \left( a(d^2) - w^s \right) \right|$$
$$\leq L_a \left| d^1 - d^2 \right| + \left| d^1 - d^2 \right| = (L_a + 1) \left| d^1 - d^2 \right|.$$

This completes the proof of Lemma C.16. □

**Proof of Lemma C.13.** This is a special case of Lemma C.4 and is hence omitted. □

## D. Proofs Omitted from Section 5

**Proof of Theorem 5.1.** We use the following lemma.

We define $h(y, x) = x \frac{e^{\beta xy}}{e^{\beta xy} + e^{\beta w^s}} y + \left( 1 - x \frac{e^{\beta xy}}{e^{\beta xy} + e^{\beta w^s}} \right) w^s$ and observe that $w_t(\mathbf{x}) = h(a(\mathcal{D}_t(\mathbf{x})), x_t)$. We analyze each property separately.

1. If $w_t^g(\mathbf{x}) \geq C$. From Proposition C.1, for every $t > \tau$ it holds that $d_t(\mathbf{x}^\tau) > d_t(\mathbf{x})$ and therefore $w_t^g(\mathbf{x}^\tau) \geq w_t^g(\mathbf{x}) \geq C$.

   Notice that

$$\frac{dh(y, x)}{dy} = x^2 \beta \frac{1}{1 + e^{\beta(w^s - xy)}} \frac{1}{1 + e^{-\beta(w^s - xy)}} (y - w^s) + x \frac{1}{1 + e^{\beta(w^s - xy)}}$$
$$\geq x^2 \beta \frac{1}{1 + e^{\beta(w^s - xy)}} \frac{1}{1 + e^{-\beta(w^s - xy)}} (xy - w^s) + x \frac{1}{1 + e^{\beta(w^s - xy)}}$$
$$\geq x^2 \beta \frac{1}{1 + e^{\beta(w^s - xy)}} \frac{1}{1 + e^{-\beta(w^s - xy)}} (xy - w^s) + x^2 \frac{1}{1 + e^{\beta(w^s - xy)}}$$
$$= x^2 \left( \beta \frac{1}{1 + e^{\beta(w^s - xy)}} \frac{1}{1 + e^{-\beta(w^s - xy)}} (xy - w^s) + \frac{1}{1 + e^{\beta(w^s - xy)}} \right)$$

Therefore, for every $x > 0$, if $\beta \frac{1}{1 + e^{\beta(w^s - xy)}} \frac{1}{1 + e^{-\beta(w^s - xy)}} (xy - w^s) + \frac{1}{1 + e^{\beta(w^s - xy)}} > 0$ then it holds that $\frac{dh(y, x)}{dy} > 0$

Next, we define the auxiliary function $g(y, x) = x \frac{e^{\beta y}}{e^{\beta y} + e^{\beta w^s}} y + \left( 1 - x \frac{e^{\beta y}}{e^{\beta y} + e^{\beta w^s}} \right) w^s$ and notice that

$$\frac{dg(y, x)}{dy} = x\beta \frac{1}{1 + e^{\beta(w^s - y)}} \frac{1}{1 + e^{-\beta(w^s - y)}} (y - w^s) + x \frac{1}{1 + e^{\beta(w^s - y)}}$$

Therefore, for every $x > 0$ it holds that

$$sign \left( \frac{dg(y, x)}{dy} \right) = sign \left( \beta \frac{1}{1 + e^{\beta(w^s - y)}} \frac{1}{1 + e^{-\beta(w^s - y)}} (y - w^s) + \frac{1}{1 + e^{\beta(w^s - y)}} \right)$$

and by definition of $g(y, x)$ we get that for every $x, y'$ such that $y = xy'$ and $\frac{dg(y, x)}{dy} > 0$ then $\frac{dh(y', x)}{dy} > 0$.

Now, we use the following lemma:

**Lemma D.1.** *For every $x > 0$ it holds that*

$$sign\left(\frac{dg(y,x)}{dy}\right) = sign(y - C).$$

Thus, according to Lemma D.1 it holds that if $w_t^g(\mathbf{x}) > C$ then $h(a(\mathcal{D}_t(\mathbf{x}^\tau)), x_t^\tau) > h(a(\mathcal{D}_t(\mathbf{x})), x_t)$ and equivalently $w_t(\mathbf{x}^\tau) > w_t(\mathbf{x})$.

2. For the next two properties, we show that there exists $y_0 \in \mathbb{R}$ such that for every $x > 0$ it holds that $\frac{dh(y_0,x)}{dy} < 0$.

First, we rewrite $\frac{dh(y,x)}{dy}$:

$$\frac{dh(y,x)}{dy} = x^2\beta\frac{1}{1+e^{\beta(w^s-xy)}}\frac{1}{1+e^{-\beta(w^s-xy)}}(y-w^s) + x\frac{1}{1+e^{\beta(w^s-xy)}}$$

$$= x\frac{1}{1+e^{\beta(w^s-xy)}}\left(1+x\beta\frac{1}{1+e^{-\beta(w^s-xy)}}(y-w^s)\right)$$

$$< x\frac{1}{1+e^{\beta(w^s-xy)}}\left(1+x\beta\frac{1}{1+e^{-\beta(w^s-xy)}}y\right)$$

$$= x\frac{1}{1+e^{\beta(w^s-xy)}}\left(1+\beta\frac{1}{1+e^{-\beta(w^s-xy)}}xy\right).$$

Notice that $x\frac{1}{1+e^{\beta(w^s-xy)}} > 0$ and therefore if $1+\beta\frac{1}{1+e^{-\beta(w^s-xy)}}xy < 0$ then $\frac{dh(y,x)}{dy} < 0$.

Next, observe that

$$\lim_{y\to-\infty} 1+\beta\frac{1}{1+e^{-\beta(w^s-xy)}}xy = \lim_{y\to-\infty} 1+\beta xy = -\infty$$

Let $\tilde{h}(yx) = 1+\beta\frac{1}{1+e^{-\beta(w^s-xy)}}xy$. Since $\frac{dh(y,x)}{dy}$ is continuous in $y$, and $h(y,x)$ is continuous and represents a lower bound of $\tilde{h}(yx)$, it holds that there exists $z_0$ such that if $yz = z_0$ then $\frac{dh(y,x)}{dy} < 0$.

Next, we use the following observation:

**Observation D.2.** *For every $x > 0$ and $y < \frac{z_0}{x}$ it holds that $\frac{dh(y,x)}{dy} < 0$.*

We denote $C' = \frac{z_0}{x}$. Using Observation D.2, if $a(\mathcal{D}_t(\mathbf{x}^\tau)) < C'$ (which means that $w_t^g(\mathbf{x}^\tau) < z_0$) then $h(a(\mathcal{D}_t(\mathbf{x}^\tau)), x_t^\tau) < h(a(\mathcal{D}_t(\mathbf{x})), x_t)$.

3. The result $h(a(\mathcal{D}_t(\mathbf{x}^\tau)), x_\tau^\tau) > h(a(\mathcal{D}_t(\mathbf{x})), x_\tau)$ follows immediately from the previous arguement.

This completes the proof of Theorem 5.1. $\qquad\square$

**Proof of Lemma D.1.** Fix $x \in [0,1]$ and we denote $\tilde{q}(y) = \frac{e^{\beta y}}{e^{\beta y}+e^{\beta w^s}}$. Therefore, $g(y)$ can be written as $g(y) = xq(y)y + (1-xq(y))w^s$. The derivative $g(y)$ is

$$\frac{dg(y)}{dy} = x\frac{d\tilde{q}(y)}{dy}y + x\tilde{q}(y) - x\frac{d\tilde{q}(y)}{dy}w^s = x\frac{d\tilde{q}(y)}{dy}(y-w^s) + x\tilde{q}(y). \qquad (5)$$

Notice that $q(y)$ is a sigmoid function and therefore $\frac{d\tilde{q}(y)}{dy} = \beta\tilde{q}(y)(1-\tilde{q}(y))$. Plugging this result in Equation 5 results in

$$\frac{dg(y)}{dy} = x\tilde{q}(y)(1-\tilde{q}(y))\beta(y-w^s) + x\tilde{q}(y).$$

Next, notice that $\tilde{q}(y) = \frac{e^{\beta y}}{e^{\beta y} + e^{\beta w^s}} = \frac{1}{1 + e^{\beta(w^s - y)}}$. We denote $z = \beta(y - w^s)$ and get

$$\begin{aligned}
\frac{dg(y)}{dy} &= x\frac{1}{1+e^{-z}}\frac{1}{1+e^z}z + x\frac{1}{1+e^{-z}} \\
&= x\frac{1}{1+e^{-z}}\left(\frac{1}{1+e^z}z + 1\right) \\
&= x\frac{1}{1+e^{-z}}\frac{z+1+e^z}{1+e^z} \\
&= x\frac{1}{1+e^{-z}}\frac{e^{z+1}}{1+e^z}\left((z+1)e^{-(z+1)} + e^{-1}\right).
\end{aligned}$$

Therefore, to find the $y_0$ that results in $\frac{dg}{dy}\big|_{y=y_0} = 0$ is equivalent to finding the solution of

$$(z+1)e^{-(z+1)} + e^{-1} = 0.$$

Denote $\tilde{z} = -(z+1)$ and we have the inverse of the Lambert function

$$\tilde{z}e^{\tilde{z}} = e^{-1}$$

and therefore $\tilde{z} = \mathcal{W}(e^{-1})$, which leads to $z_0 = -\mathcal{W}(e^{-1}) - 1$ and $y_0 = -\frac{\mathcal{W}(e^{-1})+1}{\beta} + w^s = C$.

Next, denote $h(z) = (z+1)e^{-(z+1)} + e^{-1}$ and notice that the sign of $\frac{dg}{dy}$ is determined by the sign of $h(z)$, that is $sign(\frac{dg}{dy}) = sign(h(z))$.

The derivative of $h(z)$ is given by

$$\frac{dh(z)}{dz} = e^{-(z+1)} - (z+1)e^{-(z+1)} = \left(1 - (z+1)\right)e^{-(z+1)} = -ze^{-(z+1)}.$$

Therefore $h(z)$ is an increasing function for $z < 0$ and a decreasing function for $z > 0$. Recall that $h(z_0) = 0$ and $z_0 < -1 < 0$ thus $h(z) < 0$ for every $z < z_0$. Furthermore, $h(z)$ is an increasing function in $z \in [z_0, 0)$, therefore it holds that $h(z) > 0$ for every $z \in (z_0, 0)$. Lastly, notice that for every $z > 0$ it holds that $z + 1 > 0$ and $e^{-(z+1)} > 0$. and as such we can summarize that $h(z) > 0$ for every $z > z_0$.

This completes the proof of Lemma D.1. $\qquad\square$

**Proof of Observation D.2.** Let $y^\star = \frac{z_0}{x}$. First, observe that if $\frac{dh(y^\star, x)}{dy} < 0$ then it must hold that $y^\star < w^s$. Furthermore, for any $y_1 < y^\star < w^s$ we get that

$$\begin{aligned}
0 > 1 + x\beta\frac{1}{1+e^{-\beta(w^s - xy^\star)}}(y^\star - w^s) &> 1 + x\beta\frac{1}{1+e^{-\beta(w^s - xy^\star)}}(y_1 - w^s) \\
&> 1 + x\beta\frac{1}{1+e^{-\beta(w^s - xy_1)}}(y_1 - w^s).
\end{aligned}$$

$\qquad\square$

# E. Proofs Omitted from Section 6

## E.1. Proofs Omitted from Subsection 6.2

**Proof of Corollary 6.2.** First, notice that for every $t \leq \tau$ it holds that $\mathcal{D}_t(\mathbf{x}) = \mathcal{D}_t(\mathbf{x}^\tau)$. Next, from Lemma C.8 it holds that for every $t > \tau$, the data satisfies

$$|\mathcal{D}_t(\mathbf{x}) - \mathcal{D}_t(\mathbf{x}^\tau)| \leq |\mathcal{D}_{\tau+1}(\mathbf{x}) - \mathcal{D}_{\tau+1}(\mathbf{x}^\tau)| = |p_\tau(\mathbf{x}) - p_\tau(\mathbf{x}^\tau)|.$$

Therefore, we can bound the revenue:

$$U(\mathbf{x}^\tau) - U(\mathbf{x}) \leq \gamma^{\tau-1} \left( r(p_\tau(\mathbf{x}^\tau)) - r(p_\tau(\mathbf{x})) \right) + L_r \sum_{t=\tau+1}^{T} \gamma^t \left( p_t(\mathbf{x}^\tau) - p_t(\mathbf{x}) \right).$$

By Lemma C.6 we get that

$$U(\mathbf{x}^\tau) - U(\mathbf{x}) \leq \gamma^{\tau-1} \left( r(p_\tau(\mathbf{x}^\tau)) - r(p_\tau(\mathbf{x})) \right) + L_r \sum_{t=\tau+1}^{T} \gamma^t \left( \mathcal{D}_t(\mathbf{x}^\tau) - \mathcal{D}_t(\mathbf{x}) \right)$$

$$\leq \gamma^{\tau-1} \left( r(p_\tau(\mathbf{x}^\tau)) - r(p_\tau(\mathbf{x})) \right) + L_r \left( p_\tau(\mathbf{x}) - p_\tau(\mathbf{x}^\tau) \right) \sum_{t=\tau+1}^{T} \gamma^t$$

$$\leq \gamma^{\tau-1} \left( r(p_\tau(\mathbf{x}^\tau)) - r(p_\tau(\mathbf{x})) \right) + L_r \gamma^\tau \frac{p_\tau(\mathbf{x}^\tau) - p_\tau(\mathbf{x})}{1 - \gamma}.$$

This completes the proof of Corollary 6.2. $\qquad\square$

**Theorem E.1.** *Let* $\underline{x} = \min\{x_t \mid t > \tau, x_t > 0\}$ *and* $k = \frac{\beta \min_{\mathcal{D} \in [0,T]} \frac{da(\mathcal{D})}{\mathcal{D}}}{4(1+e^{\beta w^s})^2}$. *If* $\beta L_a \leq 1$, *then*

$$U(\mathbf{x}^\tau) - U(\mathbf{x}) <$$

$$\gamma^{\tau-1} \left( r(p_\tau(\mathbf{x}^\tau)) - r(p_\tau(\mathbf{x})) \right) + L_r \gamma^\tau \frac{p_\tau(\mathbf{x}^\tau) - p_\tau(\mathbf{x})}{1 - \gamma \left( 1 - k\underline{x}^2 \right)}.$$

**Proof of Theorem E.1.** By definition, we get that

$$U(\mathbf{x}^\tau) - U(\mathbf{x}) = \sum_{t=1}^{T} \gamma^{t-1} r(p_t(\mathbf{x}^\tau)) - \sum_{t=1}^{T} \gamma^{t-1} r(p_t(\mathbf{x}))$$

$$= \sum_{t=1}^{T} \gamma^{t-1} \left( r(p_t(\mathbf{x}^\tau)) - r(p_t(\mathbf{x})) \right)$$

$$= \sum_{t=\tau}^{T} \gamma^{t-1} \left( r(p_t(\mathbf{x}^\tau)) - r(p_t(\mathbf{x})) \right)$$

$$= \gamma^{\tau-1} \left( r(p_\tau(\mathbf{x}^\tau)) - r(p_\tau(\mathbf{x})) \right) + \sum_{t=\tau+1}^{T} \gamma^{t-1} \left( r(p_t(\mathbf{x}^\tau)) - r(p_t(\mathbf{x})) \right)$$

$$\leq \gamma^{\tau-1} \left( r(p_\tau(\mathbf{x}^\tau)) - r(p_\tau(\mathbf{x})) \right) + L_r \sum_{t=\tau+1}^{T} \gamma^{t-1} \left( p_t(\mathbf{x}^\tau) - p_t(\mathbf{x}) \right).$$

Next, we use the following lemma to get an upper bound on $p_t(\mathbf{x}^\tau) - p_t(\mathbf{x})$.

**Lemma E.2.** *For every* $t > \tau$ *it holds that*

$$0 \leq p_t(\mathbf{x}^\tau) - p_t(\mathbf{x}) \leq |p_\tau(\mathbf{x}) - p_\tau(\mathbf{x}^\tau)| \prod_{i=\tau+1}^{t-1} \left( 1 - \frac{q^2}{4} x_i^2 \beta \underline{L_a} \right).$$

Therefore, according to Lemma E.2, it holds that

$$U(\mathbf{x}^\tau) - U(\mathbf{x}) \leq \gamma^{\tau-1} \left( r(p_\tau(\mathbf{x}^\tau)) - r(p_\tau(\mathbf{x})) \right) + L_r \sum_{t=\tau+1}^{T} \gamma^{t-1} |p_\tau(\mathbf{x}) - p_\tau(\mathbf{x}^\tau)| \prod_{i=\tau+1}^{t-1} \left( 1 - \frac{q^2}{4} x_i^2 \beta \underline{L_a} \right)$$

$$\leq \gamma^{\tau-1} \left( r(p_\tau(\mathbf{x}^\tau)) - r(p_\tau(\mathbf{x})) \right) + |p_\tau(\mathbf{x}) - p_\tau(\mathbf{x}^\tau)| L_r \sum_{t=\tau+1}^{T} \gamma^{t-1} x_t^2 \prod_{i=\tau+1}^{t-1} \left( 1 - \frac{q^2}{4} x_i^2 \beta \underline{L_a} \right).$$

We now simplify the second term using the following lemma.

**Lemma E.3.** *It holds that*

$$\sum_{t=\tau+1}^{T} \gamma^{t-1} x_t^2 \prod_{i=\tau+1}^{t-1} \left(1 - \frac{q^2}{4} x_i^2 \beta \underline{L_a}\right) \leq \sum_{t=\tau+1}^{T} \gamma^{t-1} \left(1 - \frac{q^2}{4} \underline{x}^2 \beta \underline{L_a}\right)^{t-\tau-1}.$$

Therefore, we get that

$$U(\mathbf{x}^\tau) - U(\mathbf{x}) \leq \gamma^{\tau-1} \left(r(p_\tau(\mathbf{x}^\tau)) - r(p_\tau(\mathbf{x}))\right) + |p_\tau(\mathbf{x}) - p_\tau(\mathbf{x}^\tau)| L_r \sum_{t=\tau+1}^{T} \gamma^{t-1} \left(1 - \frac{q^2}{4} \underline{x}^2 \beta \underline{L_a}\right)^{t-\tau-1}$$

$$= \gamma^{\tau-1} \left(r(p_\tau(\mathbf{x}^\tau)) - r(p_\tau(\mathbf{x}))\right) + |p_\tau(\mathbf{x}) - p_\tau(\mathbf{x}^\tau)| L_r \gamma^\tau \sum_{t=1}^{T-\tau} \gamma^{t-1} \left(1 - \frac{q^2}{4} \underline{x}^2 \beta \underline{L_a}\right)^{t-1}.$$

Notice that $\sum_{t=1}^{T-\tau} \gamma^{t-1} \left(1 - \frac{q^2}{4} \underline{x}^2 \beta \underline{L_a}\right)^{t-1}$ is a sum of a geometric series, and therefore it holds that

$$\sum_{t=1}^{T-\tau} \gamma^{t-1} \left(1 - \frac{q^2}{4} \underline{x}^2 \beta \underline{L_a}\right)^{t-1} = \frac{\left(\gamma \left(1 - \frac{q^2}{4} \underline{x}^2 \beta \underline{L_a}\right)\right)^{T-\tau} - 1}{\gamma \left(1 - \frac{q^2}{4} \underline{x}^2 \beta \underline{L_a}\right) - 1}.$$

Thus, we conclude that

$$U(\mathbf{x}^\tau) - U(\mathbf{x}) \leq \gamma^{\tau-1} \left(r(p_\tau(\mathbf{x}^\tau)) - r(p_\tau(\mathbf{x}))\right) + |p_\tau(\mathbf{x}) - p_\tau(\mathbf{x}^\tau)| L_r \gamma^\tau \frac{\left(\gamma \left(1 - \frac{q^2}{4} \underline{x}^2 \beta \underline{L_a}\right)\right)^{T-\tau} - 1}{\gamma \left(1 - \frac{q^2}{4} \underline{x}^2 \beta \underline{L_a}\right) - 1}.$$

This completes the proof of Theorem E.1. □

**Proof of Lemma E.2.** We start from the left inequality. From Theorem 4.1 it holds that $p_t(\mathbf{x}^\tau) \geq p_t(\mathbf{x})$ for every $t > \tau$. We move on to the right inequality. For that, we use Lemma C.6 and get that

$$p_t(\mathbf{x}^\tau) - p_t(\mathbf{x}) < \mathcal{D}_t(\mathbf{x}^\tau) - \mathcal{D}_t(\mathbf{x}).$$

Next, we couple it with the following lemma.

**Lemma E.4.** *For every $t > \tau$ it holds that*

$$0 < \mathcal{D}_t(\mathbf{x}^\tau) - \mathcal{D}_t(\mathbf{x}) \leq |p_\tau(\mathbf{x}) - p_\tau(\mathbf{x}^\tau)| \prod_{i=\tau+1}^{t-1} \left(1 - \frac{q^2}{4} x_i^2 \beta \underline{L_a}\right).$$

Therefore, we conclude that

$$p_t(\mathbf{x}^\tau) - p_t(\mathbf{x}) < \mathcal{D}_t(\mathbf{x}^\tau) - \mathcal{D}_t(\mathbf{x})$$

$$\leq |p_\tau(\mathbf{x}) - p_\tau(\mathbf{x}^\tau)| \prod_{i=\tau+1}^{t-1} \left(1 - \frac{q^2}{4} x_i^2 \beta \underline{L_a}\right).$$

This completes the proof of Lemma E.2. □

**Proof of Lemma E.7.** We expand it according to the definition:

$$\left| q(\mathcal{D}^1, x^1) - q(\mathcal{D}^2, x^2) \right| = \left| \frac{e^{\beta a(\mathcal{D}^1)x^1}}{e^{\beta a(\mathcal{D}^1)x^1} + e^{\beta w^s}} - \frac{e^{\beta a(\mathcal{D}^2)x^2}}{e^{\beta a(\mathcal{D}^2)x^2} + e^{\beta w^s}} \right|$$

$$= \left| \frac{e^{\beta a(\mathcal{D}^1)x^1}\left(e^{\beta a(\mathcal{D}^2)x^2} + e^{\beta w^s}\right) - e^{\beta a(\mathcal{D}^2)x^2}\left(e^{\beta a(\mathcal{D}^1)x^1} + e^{\beta w^s}\right)}{\left(e^{\beta a(\mathcal{D}^1)x^1} + e^{\beta w^s}\right)\left(e^{\beta a(\mathcal{D}^2)x^2} + e^{\beta w^s}\right)} \right|$$

$$= \left| e^{\beta w^s} \frac{e^{\beta a(\mathcal{D}^1)x^1} - e^{\beta a(\mathcal{D}^2)x^2}}{\left(e^{\beta a(\mathcal{D}^1)x^1} + e^{\beta w^s}\right)\left(e^{\beta a(\mathcal{D}^2)x^2} + e^{\beta w^s}\right)} \right|$$

$$= \left| e^{\beta w^s} e^{\beta a(\mathcal{D}^1)x^1} \frac{1 - e^{\beta\left(x^2 a(\mathcal{D}^2) - x^1 a(\mathcal{D}^1)\right)}}{\left(e^{\beta a(\mathcal{D}^1)x^1} + e^{\beta w^s}\right)\left(e^{\beta a(\mathcal{D}^2)x^2} + e^{\beta w^s}\right)} \right|$$

$$= \left| q(\mathcal{D}^1, x^1)\left(1 - q(\mathcal{D}^2, x^2)\right)\left(1 - e^{\beta\left(x^2 a(\mathcal{D}^2) - x^1 a(\mathcal{D}^1)\right)}\right) \right|$$

$$= q(\mathcal{D}^1, x^1)\left(1 - q(\mathcal{D}^2, x^2)\right)\left| 1 - e^{\beta\left(x^2 a(\mathcal{D}^2) - x^1 a(\mathcal{D}^1)\right)} \right|.$$

This completes the proof of Lemma E.7. $\qquad \square$

**Proof of Lemma E.4.** We prove it by induction, starting with the base case at $t = \tau + 1$. By definition,

$$\left| \mathcal{D}_{\tau+1}(\mathbf{x}) - \mathcal{D}_{\tau+1}(\mathbf{x}^\tau) \right| = \left| \mathcal{D}_\tau(\mathbf{x}) - p_\tau(\mathbf{x}) - \mathcal{D}_\tau(\mathbf{x}^\tau) + p_\tau(\mathbf{x}^\tau) \right|.$$

Since $\mathcal{D}_\tau(\mathbf{x}) = \mathcal{D}_\tau(\mathbf{x}^\tau)$ we get that

$$\left| \mathcal{D}_{\tau+1}(\mathbf{x}) - \mathcal{D}_{\tau+1}(\mathbf{x}^\tau) \right| = \left| p_\tau(\mathbf{x}) - p_\tau(\mathbf{x}^\tau) \right|.$$

Therefore, we can conclude the base case. Next, assume that the inequality holds for $t > \tau + 1$, and we prove for $t + 1$. We use the following lemma:

**Lemma E.5.** *For every $t > \tau + 1$ it holds that*

$$\left| \mathcal{D}_{t+1}(\mathbf{x}) - \mathcal{D}_{t+1}(\mathbf{x}^\tau) \right| \leq \left( 1 - \frac{q^2}{4} x_t^2 \beta \underline{L_a} \right) \left| \mathcal{D}_t(\mathbf{x}) - \mathcal{D}_t(\mathbf{x}^\tau) \right|.$$

We plug the inequality from our assumption into the inequality of lemma E.5, Therefore, we get that

$$\left| \mathcal{D}_{t+1}(\mathbf{x}) - \mathcal{D}_{t+1}(\mathbf{x}^\tau) \right| \leq \left( 1 - \frac{q^2}{4} x_t^2 \beta \underline{L_a} \right) \left| \mathcal{D}_t(\mathbf{x}) - \mathcal{D}_t(\mathbf{x}^\tau) \right|$$

$$= \left( 1 - \frac{q^2}{4} x_t^2 \beta \underline{L_a} \right) \left( \mathcal{D}_t(\mathbf{x}^\tau) - \mathcal{D}_t(\mathbf{x}) \right)$$

$$\leq \left( 1 - \frac{q^2}{4} x_t^2 \beta \underline{L_a} \right) \left| p_\tau(\mathbf{x}) - p_\tau(\mathbf{x}^\tau) \right| \prod_{i=\tau+1}^{t-1} \left( 1 - \frac{q^2}{4} x_i^2 \beta \underline{L_a} \right)$$

$$= \left| p_\tau(\mathbf{x}) - p_\tau(\mathbf{x}^\tau) \right| \prod_{i=\tau+1}^{t} \left( 1 - \frac{q^2}{4} x_i^2 \beta \underline{L_a} \right).$$

This completes the proof of Lemma E.4. $\qquad \square$

**Proof of Lemma E.5.** By definition,

$$|\mathcal{D}_{t+1}(\mathbf{x}) - \mathcal{D}_{t+1}(\mathbf{x}^\tau)| = |\mathcal{D}_t(\mathbf{x}) - \mathcal{D}_t(\mathbf{x}^\tau) + p_t(\mathbf{x}^\tau) - p_t(\mathbf{x})|.$$

Since $y < x_\tau$ then from Theorem 4.1 it holds that $\mathcal{D}_t(\mathbf{x}^\tau) > \mathcal{D}_t(\mathbf{x})$ and $p_t(\mathbf{x}^\tau) > p_t(\mathbf{x})$ for every $t > \tau$. Next, we get an upper bound using the following lemma, which suggests a lower bound for the proportions.

**Lemma E.6.** *For every $t > \tau$, it holds that*

$$q(\mathcal{D}_t(\mathbf{x}^\tau), x_t) - q(\mathcal{D}_t(\mathbf{x}), x_t) \geq \frac{q^2}{4} x\beta\underline{L_a}\left(\mathcal{D}_t(\mathbf{x}^\tau) - \mathcal{D}_t(\mathbf{x})\right).$$

Using Lemma E.6, we get that

$$\begin{aligned}
|\mathcal{D}_{t+1}(\mathbf{x}) - \mathcal{D}_{t+1}(\mathbf{x}^\tau)| &= \mathcal{D}_t(\mathbf{x}^\tau) - \mathcal{D}_t(\mathbf{x}) + p_t(\mathbf{x}) - p_t(\mathbf{x}^\tau) \\
&\leq \mathcal{D}_t(\mathbf{x}^\tau) - \mathcal{D}_t(\mathbf{x}) - \frac{q^2}{4} x_t^2 \beta\underline{L_a}\left(\mathcal{D}_t(\mathbf{x}^\tau) - \mathcal{D}_t(\mathbf{x})\right) \\
&= (\mathcal{D}_t(\mathbf{x}^\tau) - \mathcal{D}_t(\mathbf{x}))\left(1 - \frac{q^2}{4} x_t^2 \beta\underline{L_a}\right) \\
&= \left(1 - \frac{q^2}{4} x_t^2 \beta\underline{L_a}\right)|\mathcal{D}_t(\mathbf{x}) - \mathcal{D}_t(\mathbf{x}^\tau)|.
\end{aligned}$$

This completes the proof of Lemma E.5. $\square$

**Proof of Lemma E.6.** From Theorem 4.1, for every $t > \tau$ it holds that $\mathcal{D}_t(\mathbf{x}^\tau) > \mathcal{D}_t(\mathbf{x})$. Therefore, we get that $a(\mathcal{D}_t(\mathbf{x}^\tau)) > a(\mathcal{D}_t(\mathbf{x}))$. Furthermore, from Proposition C.2 it holds that $q(\mathcal{D}_t(\mathbf{x}^\tau), x_t) \geq q(\mathcal{D}_t(\mathbf{x}), x_t)$. Thus, we use the following lemma to write $q(\mathcal{D}_t(\mathbf{x}^\tau), x) - q(\mathcal{D}_t(\mathbf{x}), x)$ differently:

**Lemma E.7.** *For every $\mathcal{D}^1, \mathcal{D}^2 \in [0, T]$ and $x^1, x^2 \in [0, 1]$ it holds that*

$$\left|q(\mathcal{D}^1, x^1) - q(\mathcal{D}^2, x^2)\right| = q(\mathcal{D}^1, x^1)\left(1 - q(\mathcal{D}^2, x^2)\right)\left|1 - e^{\beta\left(x^2 a(\mathcal{D}^2) - x^1 a(\mathcal{D}^1)\right)}\right|.$$

Therefore,

$$\begin{aligned}
q(\mathcal{D}_t(\mathbf{x}^\tau), x) - q(\mathcal{D}_t(\mathbf{x}), x) &= |q(\mathcal{D}_t(\mathbf{x}^\tau), x) - q(\mathcal{D}_t(\mathbf{x}), x)| \\
&= q(\mathcal{D}_t(\mathbf{x}^\tau), x)\left(1 - q(\mathcal{D}_t(\mathbf{x}), x)\right)\left|1 - e^{x\beta(a(\mathcal{D}_t(\mathbf{x})) - a(\mathcal{D}_t(\mathbf{x}^\tau)))}\right|.
\end{aligned} \tag{6}$$

Notice that $q(\mathcal{D}, x), 1 - q(\mathcal{D}, x) \geq \underline{q}$ for every $\mathcal{D} \in [0, T]$ and $x \in [0, 1]$. Furthermore, it holds that $\left|a(\mathcal{D}^2) - a(\mathcal{D}^1)\right| \geq \underline{L_a}\left|\mathcal{D}^2 - \mathcal{D}^1\right|$. Therefore,

$$a(\mathcal{D}_t(\mathbf{x})) - a(\mathcal{D}_t(\mathbf{x}^\tau)) = -|a(\mathcal{D}_t(\mathbf{x})) - a(\mathcal{D}_t(\mathbf{x}^\tau))| \leq -\underline{L_a}|\mathcal{D}_t(\mathbf{x}) - \mathcal{D}_t(\mathbf{x}^\tau)| = \underline{L_a}(\mathcal{D}_t(\mathbf{x}) - \mathcal{D}_t(\mathbf{x}^\tau)).$$

Notice that $\underline{L_a}(\mathcal{D}_t(\mathbf{x}) - \mathcal{D}_t(\mathbf{x}^\tau)) \leq 0$ and therefore $\left|1 - e^{x\beta(a(\mathcal{D}_t(\mathbf{x})) - a(\mathcal{D}_t(\mathbf{x}^\tau)))}\right| > \left|1 - e^{x\beta\underline{L_a}(\mathcal{D}_t(\mathbf{x}) - \mathcal{D}_t(\mathbf{x}^\tau))}\right|$. Plugging everything into Equation (6) results in the following inequality:

$$q(\mathcal{D}_t(\mathbf{x}^\tau), x) - q(\mathcal{D}_t(\mathbf{x}), x) \geq \underline{q}^2\left|1 - e^{x\beta\underline{L_a}(\mathcal{D}_t(\mathbf{x}) - \mathcal{D}_t(\mathbf{x}^\tau))}\right|.$$

Next, we show that $x\beta\underline{L_a}|\mathcal{D}_t(\mathbf{x}) - \mathcal{D}_t(\mathbf{x}^\tau)| \leq 1$. For that, we use the following lemma.

**Lemma E.8.** *For every $t > \tau$ it holds that $|\mathcal{D}_t(\mathbf{x}) - \mathcal{D}_t(\mathbf{x}^\tau)| \leq 1$.*

Therefore, we get that

$$x\beta\underline{L_a}\,|\mathcal{D}_t(\mathbf{x}) - \mathcal{D}_t(\mathbf{x}^\tau)| \le x\beta\underline{L_a} \le \beta\underline{L_a} \le \beta L_a \le 1.$$

Thus, we can use the inequality $|1 - e^\alpha| \ge \frac{|\alpha|}{4}$ for $|\alpha| \le 1$ and conclude that

$$q(\mathcal{D}_t(\mathbf{x}^\tau), x) - q(\mathcal{D}_t(\mathbf{x}), x) \ge \frac{q^2}{4}x\beta\underline{L_a}\,|\mathcal{D}_t(\mathbf{x}) - \mathcal{D}_t(\mathbf{x}^\tau)|$$
$$\frac{q^2}{4}x\beta\underline{L_a}\,(\mathcal{D}_t(\mathbf{x}^\tau) - \mathcal{D}_t(\mathbf{x})).$$

This completes the proof of Lemma E.6. $\qquad\square$

**Proof of Lemma E.8.** By definition, we get that

$$
\begin{aligned}
|\mathcal{D}_t(\mathbf{x}) - \mathcal{D}_t(\mathbf{x}^\tau)| &= |\mathcal{D}_{t-1}(\mathbf{x}) + (1 - p_{t-1}(\mathbf{x})) - \mathcal{D}_{t-1}(\mathbf{x}^\tau) - (1 - p_{t-1}(\mathbf{x}^\tau))| \\
&= |\mathcal{D}_{t-1}(\mathbf{x}) - p_{t-1}(\mathbf{x}) - \mathcal{D}_{t-1}(\mathbf{x}^\tau) + p_{t-1}(\mathbf{x}^\tau)| \\
&= \mathcal{D}_{t-1}(\mathbf{x}^\tau) - \mathcal{D}_{t-1}(\mathbf{x}) + p_{t-1}(\mathbf{x}) - p_{t-1}(\mathbf{x}^\tau).
\end{aligned}
$$

Observe that the proportions satisfies that $p_{t-1}(\mathbf{x}) - p_{t-1}(\mathbf{x}^\tau) \le 0$. Therefore,

$$|\mathcal{D}_t(\mathbf{x}) - \mathcal{D}_t(\mathbf{x}^\tau)| \le |\mathcal{D}_{t-1}(\mathbf{x}^\tau) - \mathcal{D}_{t-1}(\mathbf{x})|\,.$$

Thus, by induction it follows that

$$
\begin{aligned}
|\mathcal{D}_t(\mathbf{x}) - \mathcal{D}_t(\mathbf{x}^\tau)| &\le |\mathcal{D}_{\tau+1}(\mathbf{x}) - \mathcal{D}_{\tau+1}(\mathbf{x}^\tau)| \\
&= |\mathcal{D}_\tau(\mathbf{x}) + (1 - p_\tau(\mathbf{x})) - \mathcal{D}_\tau(\mathbf{x}^\tau) - (1 - p_\tau(\mathbf{x}^\tau))| \\
&= |p_\tau(\mathbf{x}^\tau) - p_\tau(\mathbf{x})| \le 1.
\end{aligned}
$$

This completes the proof of Lemma E.8. $\qquad\square$

**Proof of Lemma E.3.** Let $t' > \tau$ be the maximum $t' \in [T]$ such that $x_{t'} = 0$. Therefore,

$$
\begin{aligned}
&\sum_{t=\tau+1}^{T} \gamma^{t-1} x_t^2 \prod_{i=\tau+1}^{t-1} \left(1 - \frac{q^2}{4}x_i^2\beta\underline{L_a}\right) \\
&= \sum_{t=\tau+1}^{t'-1} \gamma^{t-1} x_t^2 \prod_{i=\tau+1}^{t-1} \left(1 - \frac{q^2}{4}x_i^2\beta\underline{L_a}\right) + \sum_{t=t'+1}^{T} \gamma^{t-1} x_t^2 \prod_{i=\tau+1}^{t-1} \left(1 - \frac{q^2}{4}x_i^2\beta\underline{L_a}\right) \\
&\le \sum_{t=\tau+1}^{t'-1} \gamma^{t-1} x_t^2 \prod_{i=\tau+1}^{t-1} \left(1 - \frac{q^2}{4}x_i^2\beta\underline{L_a}\right) + \sum_{t=t'+1}^{T} \gamma^{t-1} \prod_{i=\tau+1}^{t-1} \left(1 - \frac{q^2}{4}x_i^2\beta\underline{L_a}\right).
\end{aligned}
$$

We now focus on the second term:

$$
\sum_{t=t'+1}^{T} \gamma^{t-1} \prod_{i=\tau+1}^{t-1} \left(1 - \frac{q^2}{4} x_i^2 \beta \underline{L_a}\right) = \sum_{t=t'+1}^{T} \gamma^{t-1} \prod_{\substack{i=\tau+1 \\ i \neq t'}}^{t-1} \left(1 - \frac{q^2}{4} x_i^2 \beta \underline{L_a}\right)
$$

$$
\leq \sum_{t=t'+1}^{T} \gamma^{t-1} \prod_{\substack{i=\tau+1 \\ i \neq t'}}^{t-1} \left(1 - \frac{q^2}{4} \underline{x}^2 \beta \underline{L_a}\right)
$$

$$
= \sum_{t=t'+1}^{T} \gamma^{t-1} \left(1 - \frac{q^2}{4} \underline{x}^2 \beta \underline{L_a}\right)^{t-\tau-2}
$$

$$
= \sum_{t=t'}^{T-1} \gamma^{t} \left(1 - \frac{q^2}{4} \underline{x}^2 \beta \underline{L_a}\right)^{t-\tau-1}
$$

$$
\leq \sum_{t=t'}^{T-1} \gamma^{t-1} \left(1 - \frac{q^2}{4} \underline{x}^2 \beta \underline{L_a}\right)^{t-\tau-1}
$$

$$
\leq \sum_{t=t'}^{T} \gamma^{t-1} \left(1 - \frac{q^2}{4} \underline{x}^2 \beta \underline{L_a}\right)^{t-\tau-1}.
$$

Therefore, we conclude that

$$
\sum_{t=\tau+1}^{T} \gamma^{t-1} x_t^2 \prod_{i=\tau+1}^{t-1} \left(1 - \frac{q^2}{4} x_i^2 \beta \underline{L_a}\right) \leq \sum_{t=\tau+1}^{t'-1} \gamma^{t-1} x_t^2 \prod_{i=\tau+1}^{t-1} \left(1 - \frac{q^2}{4} x_i^2 \beta \underline{L_a}\right) + \sum_{t=t'}^{T} \gamma^{t-1} \left(1 - \frac{q^2}{4} \underline{x}^2 \beta \underline{L_a}\right)^{t-\tau-1}.
$$

At this point, we iteratively apply it while going backward using backward induction. In each step, we take the latest round $t'$ such that $x_t' = 0$ and apply the equation above to the first term. Ultimately, we get that

$$
\sum_{t=\tau+1}^{T} \gamma^{t-1} x_t^2 \prod_{i=\tau+1}^{t-1} \left(1 - \frac{q^2}{4} x_i^2 \beta \underline{L_a}\right) \leq \sum_{t=\tau+1}^{T} \gamma^{t-1} \left(1 - \frac{q^2}{4} \underline{x}^2 \beta \underline{L_a}\right)^{t-\tau-1}.
$$

This completes the proof of Lemma E.3. □

## F. Simulations

In this section we provide empirical demonstration of the effect of selective responses.

**Experimental Model**    We set our model with the following parameters: $T = 50, w^s = 0.5, a(\mathcal{D}) = 1 - e^{-4\mathcal{D}}$ and $r(p) = p^\alpha$ for $\alpha \in \mathbb{R}$. We show our results for varying values of $\beta, \gamma$ and $\alpha$. Ideally, we would compute the optimal selective response strategy for each set of model parameters, but doing so requires using a fine discretization of the range $[0, 1]$ for each selection of $x_t$. To ease our computation for large values of $T$, we choose the optimal selective response strategy in the set of *cutoff strategies*. We denote by $X_c$ the set of cutoff strategies, namely, the set of all strategies in which GenAI remains does not answer until a specific round and fully responds thereafter. Formally, for every strategy $\mathbf{x} \in X_c$, there exists $\tau \in T$ such that $x_t = 0$ for all $t < \tau$ and $x_t = 1$ for all $t \geq \tau$. For each set of model parameters, we calculate the optimal cutoff strategy to maximize GenAI's revenue and plot the differences in GenAI revenue and the users social welfare relative to the revenue and welfare induced by the full response strategy.

**Experiment Setup**    We report the induced revenues and welfare from 60 instances. We used a standard PC with intel Core i7-9700k CPU and 16GB RAM for running the simulations. The entire execution took roughly 1 hour.

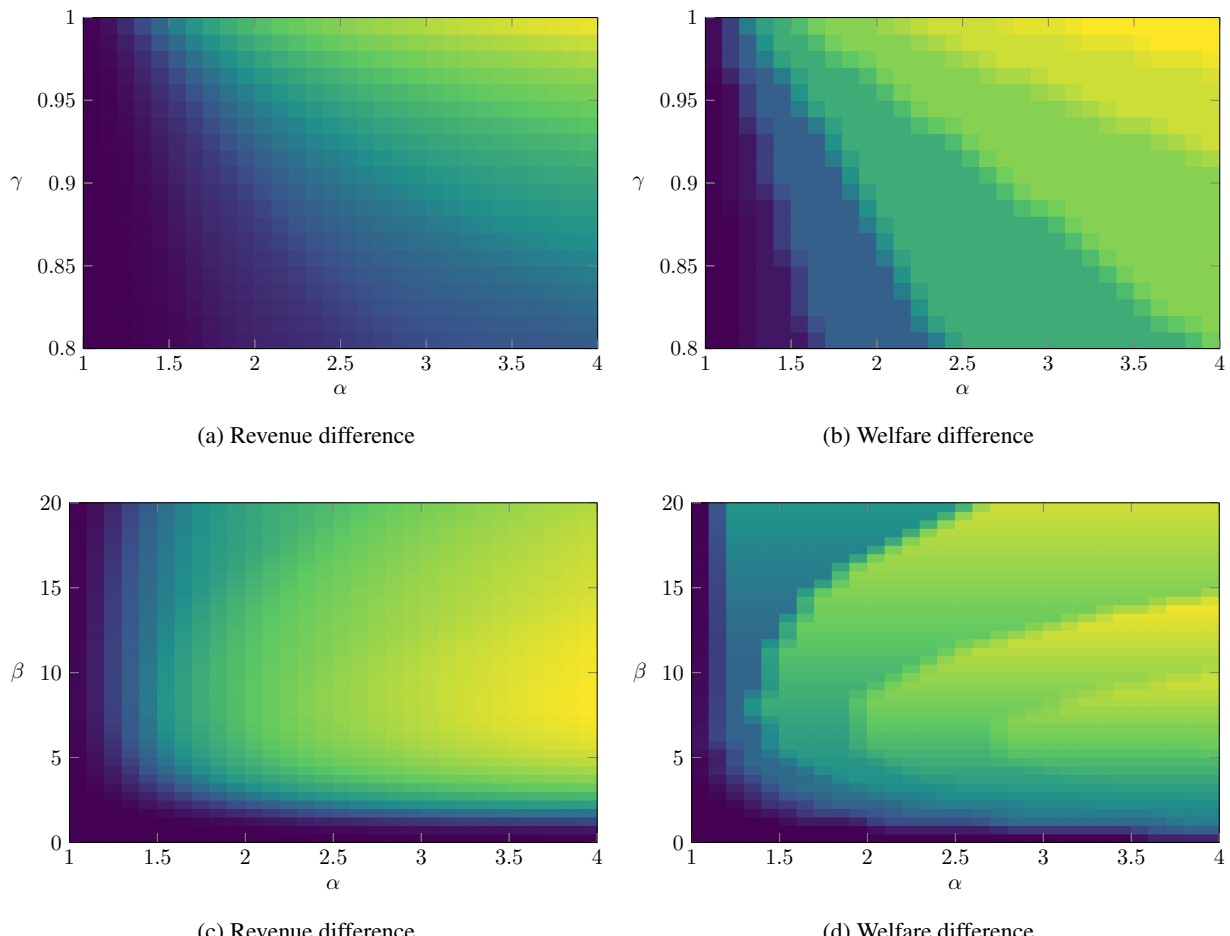

(a) Revenue difference

(b) Welfare difference

(c) Revenue difference

(d) Welfare difference

Figure 4: Revenue and welfare difference between the GenAI's optimal cutoff selective response and the full response strategy as a function of the revenue scaling power $\alpha$ and temperature $\beta$.

### F.1. Results

Figure 4 illustrates how GenAI's revenue (users' social welfare) changes with respect to the discount paramter $\gamma$, the proportions scaling power $\alpha$ and the sensitivity parameter $\beta$. Light colors indicate high differences in the revenue and social welfare, while darker colors indicate lower differences. Figure 4a and Figure 4c shows the difference between the optimal cutoff strategy and the full response strategy in log scale. Formally $\max_{\mathbf{x} \in X_c} U(\mathbf{x}) - U(\bar{\mathbf{x}})$. Figure 4b and Figure 4d illustrate the difference in the users' social welfare between GenAI's optimal cutoff strategy and the full-response strategy. Formally we compute $W(\mathbf{x}) - W(\bar{\mathbf{x}})$ such that $\mathbf{x} \in \max_{\mathbf{x} \in X_c} U(\mathbf{x})$.

In our first experiment, we set $\beta = 5$ and computed the revenue and social welfare for various combinations of $\alpha$ and $\gamma$. As shown in Figure 4a, revenue increases with both $\alpha$ and $\gamma$. Observe that decreasing the discount factor $\gamma$ and increasing the power $\alpha$ can have opposing effects on revenue. A lower $\gamma$ makes GenAI more myopic, favoring strategies that maximize immediate gains from the current proportions. Consequently, it tends to adopt strategies close to full response, resulting in minor revenue differences. In contrast, a higher $\alpha$ amplifies the influence of high proportions on revenue. This drives GenAI to prioritize increasing the proportions, even at the cost of short-term revenue, leading more to a long-term approach.

Figure 4b presents the social welfare as a function of $\alpha$ and $\gamma$. Although users' welfare does not explicitly depend on GenAI's discount factor $\gamma$, we observe six distinct regions with uniform color. This dependence arises through GenAI's strategy, which is sensitive to $\gamma$. Within each region, GenAI adopts the same selective response strategy across all combinations of $\alpha$ and $\gamma$. As discussed in Section 5, using a selective response when GenAI is inaccurate improves social welfare. This typically occurs for small values of $t$, when GenAI prioritizes future proportions, which is the same underlying reason

behind the observed increase in revenue as $\alpha$ and $\gamma$ grow.

Figures 4c and 4d show the differences in revenue and welfare as functions of the temperature $\beta$ and the power parameter $\alpha$. When $\beta = 0$, users are indifferent between platforms, regardless of the utilities they receive. In this case, the accuracy of GenAI, and therefore the data it accumulates, has no impact on user decisions. As a result, choosing a selective response can only reduce both GenAI's revenue and social welfare.

As $\beta$ increases, users become more sensitive to utility differences, and the value of the data GenAI accumulates becomes more apparent. This leads to two opposing effects. First, selective response allows GenAI to influence the amount of data generated in each round, which in turn affects how many users choose GenAI. Second, when $\beta$ becomes very large, the effectiveness of selective response decreases. In this case, user behavior resembles a best response dynamic: as long as GenAI's utility is lower than that of Forum, users prefer Forum and generate data there, regardless of GenAI's strategy. Once GenAI's utility exceeds that of Forum, the optimal strategy is to always choose the full response, since users will consistently choose GenAI.

Notably, the welfare in Figure 4d exhibits the same pattern as the revenue, but in a more nuanced fashion. This is because welfare is influenced by both GenAI's response strategy and $\beta$. From Theorem 5.1, it follows that the threshold $C$ increases with $\beta$, and therefore using a selective response when the utility from GenAI is below $C$ may lead to a decrease in welfare. This results in a cyclic pattern in welfare when $\alpha$ is held constant.

