# OpenReview forum: "Selective Response Strategies for GenAI"
_ICML.cc/2025/Conference — ICML 2025 poster_

### Official Review · Reviewer_S9Pd · 2025-03-08

**Overall Recommendation:** 4

**Summary:**

This paper introduces the concept of "selective response" for GenAI systems.

Based on this notion, the contribution of the paper is two-fold.

First, on the conceptual side, it  presents a stylized model with two platforms (a GenAI system and a human-driven forum) where users choose between them sequentially. In this model, by sometimes not responding optimally, GenAI systems drive users to human forums like Stack Overflow, generating valuable training data that later improves the GenAI system.

On the technical side, the paper shows that under this model,  selective response can improve both GenAI revenue, user welfare, or even both, compared to always-responding. Finally, the work provides an approximately optimal algorithm for maximizing GenAI revenue under social welfare constraints.

**Claims And Evidence:**

The main results of the work are theoretical.  In particular, the paper shows that

* Selective response can Pareto-dominate always-responding strategies (Observation 3.1)
* The revenue inefficiency of always responding can be unbounded (Proposition 3.2)
* Selective response can nearly double social welfare in some instances (Proposition 3.3)
* For any strategy, a selective modification generates more future data (Theorem 4.1)

The claims are primarily theoretical, with formal proofs. I have checked the main proof of Theorem 1.1 and 1.2, and the proofs are sound.

At a conceptual level, the insight from this paper challenges the conventional notion that
GenAI should always provide answers. I find that this idea is well supported by the theoretical claims.

(It's not a weakness of the work at all, but note that the paper is not experimental and provides no empirical benchmark to evaluate their results.)

**Essential References Not Discussed:**

I am not familiar with the literature on game theory

**Experimental Designs Or Analyses:**

The paper is primarily theoretical and does not include empirical experiments. While this is acceptable for a foundational theoretical paper, even simple simulations would have strengthened the work. Specifically, I wonder if the dynamics can be simulated by multiple self-training rounds on synthetic data with simpler models.

**Methods And Evaluation Criteria:**

The paper uses a game theoretical model to study the dynamics between GenAI and human responses on question-answer platforms.  I think the lens of game theory is appropriate for analyzing strategic interactions between platforms and users.

Under this model, the paper provides a theoretical guarantee on selective response and inspired by that, an approximately optimal algorithm for social welfare.

While the theoretical results, in my view, are strong and significant, they are somewhat simplified relative to the real-world GenAI dynamics. For example, as the author(s) admitted, the paper only considers the case where a single GenAI platform exists. Incooporating the full dynamics of multiple GenAI platforms and/or modelling multiple heterogeneous user distributions could make the work stronger.

**Other Comments Or Suggestions:**

na

**Other Strengths And Weaknesses:**

Overall, I find the paper very well written and clearly structured.

The conceptual message was delivered and supported by the theoretical claims. Importantly, I think the regulatory perspective provides actionable insights for policymakers (section 6), where they give conditions under which policy interventions improve welfare.

To my knowledge, the main theoretical framework of the paper is novel (though, again, I am not an expert in game-theory or the study of AI and society).

As the author(s) acknowledged, the paper does make a few simplifying assumptions, although, in my view, this is a good work as the first step in studying this dynamics between GenAI and human responses. Finally, I believe the paper could be strengthened if the author(s) could validate their results with a bit of experimental simulation.

Given the strength of the theoretical results and novelty of the main framework, I lean towards accept.

**Questions For Authors:**

Have you considered how users might react if they become aware of selective response strategies from GenAI? Also, do we assume that, when posting contents, the GenAI always reveal their identity, if that matters at all under this model?

**Relation To Broader Scientific Literature:**

I am not familiar with the literature on game theory

**Theoretical Claims:**

I reviewed several of the key theoretical claims, mostly surroudning theorem 1.1 and 1.2. The proofs appear sound.

---

> ### Author Rebuttal · Authors · 2025-03-28
>
> We thank the reviewer for the thoughtful evaluation and for finding our work novel.
>
> Addressing the reviewer's questions:
>
> 1. Regarding "*how users might react if they become aware of selective response strategies from GenAI?*"
> While we model users as non-strategic and indifferent to the actions of GenAI, we totally agree that this might be a concern. There are many ways to reconcile it:
>
>     - This behavior is akin to multi-armed bandit settings, in which to maximize the total reward, we sometimes have to explore (trading short-term reward for future rewards). Users that participate in exploration rounds effectively receive a lower utility from the system. We cannot argue that users will comply with this behavior for the GenAI case, but such multi-armed bandit mechanisms are prevalent in real-world settings.
>
>     - If selective response is used in a way that is opaque to users, users would not be able to tell. Thus, we can assume they are indifferent to it. Alternatively, if GenAI is transparent about its need for more data on a matter, a message that could be conveyed by saying it has low confidence, users might respond allostratically.
>
>     - Future work could explicitly model user dissatisfaction of deliberate selective response, potentially optimizing revenue, welfare, or other metrics under more complex user behavior.
>
>     This comment is spot-on, and, upon acceptance, we intend to use the extra space to extend this discussion. Thanks for raising it.
>
>
> 2. Regarding "*do we assume that, when posting contents, the GenAI always reveal their identity, if that matters at all under this model?*"  We do not make any assumptions regarding GenAI revealing user identities. Moreover, GenAI's actions are independent of user identities.
>
> The reviewer also mentions that studying "*the full dynamics of multiple GenAI platforms… could make the work stronger.*" Moving from one GenAI to several requires a more game-theoretic approach; we see this as a promising future work. Our intuition is that we would get a Prisoner's Dilemma-like interaction. If one platform chooses to withhold responses, it benefits all other platforms while preserving the same data generation dynamics as always responding. Conversely, if all platforms choose to withhold, they would collectively be better off. In this scenario, the payoffs (expected revenue) depend on the decisions of all platforms as well as user preferences. As in the classic Prisoner's Dilemma, collective cooperation (i.e., not answering) could be enforced through contracts or incentivized by other means.

---

> > ### Comment · Reviewer_S9Pd · 2025-04-07
> >
> > Thanks for clarifying my questions! I look forward to a future work on multi-platform setting.
> >
> > I raised my rating and would recommend accept.

---

### Official Review · Reviewer_KDdP · 2025-03-13

**Overall Recommendation:** 5

**Summary:**

This paper contributes a framework for optimizing model generation for data generation  (e.g., encouraging engagement on a forum) and long-term revenue rather than for quality and completeness of response. The authors adopt a two party model where a generative model and a forum share a user set. Over a series of time steps, users decide between GenAI and Forum by comparing the expected utility they derive from each platform. Notably, the GenAI utility is defined by a cumulative data function while the Forum utility is constant. They then show the always-responding condition is Pareto-dominated by selective response, contrasting social welfare and GenAI’s revenue.

The authors' model closely relates to economics papers like Bergemann & Bonatti (2024) and McIntyre & Srinivasan (2017), which consider how platforms & data accumulation, shape user experiences and firm outcomes. To this end, the contributions are specifically a framing of how selectively *not* responding may benefit the overall ecosystem.

**Claims And Evidence:**

1. This work is a purely theoretical paper with no experimental data, simulations, or case studies to validate the claim that selective response increases data quality or user welfare. The work falls closer to traditional economics than representative learning, but it is topically of great interest to the ICML community.

2. The paper does what it claims, with a number of assumptions that introduce limitations or constraints on its usefulness. For example, the assumption that Forum utility can be treated as a constant is surprising through perhaps reasonable simplification. Forum utility can also be a function of cumulative content or user engagement, which may add complexity to the analysis but benefits the GenAI system (GenAI gets more training data for new content, at a compounding rate)--and can be done without extending into two-player strategic games. The concept of “not responding” is homogenized, but in practice there are substantive differences between refusal, incorrect information, and redirection (sending users to the Forum), each of which would have different effects on user experience, and similarly offer different penalties. These simplifications are largely fine, however, given this is an idealized model.

3. The softmax definition of user choice is symmetric and memoryless (though includes a sensitivity parameter that captures users’ responsiveness to utility differences). It doesn’t account for any of the following effects:
(1) User stickiness (repeated use of the same platform)
(2) Frustration penalty if GenAI gives poor/no answers
(3) Platform trust (e.g., if GenAI’s quality fluctuates, do users downgrade expectations?)
all of which are important context for the argument that GenAI platforms benefit from refusal / incomplete responses. To be more concrete,  single-round decisions are a significant simplification.

**Essential References Not Discussed:**

The related works are largely covered within the paper. The authors may be interested in extended discussion contrasting generated data with human-produced data, such as Shumailov et al. (2023), “The Curse of Recursion: Training on Generated Data Makes Models Forget" as further motivation for emphasizing human data production.

**Experimental Designs Or Analyses:**

There are none.

**Methods And Evaluation Criteria:**

There is no substantive evaluation as this is a theoretical paper.

**Other Comments Or Suggestions:**

The regulator's capabilities are barely sketched. It would be nice to include further discussion on this point by the authors.

*** I have updated my score to reflect the authors continued rebuttal engagement and addition of simulation.

**Other Strengths And Weaknesses:**

The setting is novel and of interest to the community. The paper is generally well-written. There are a number of assumptions that introduce some weaknesses to the work. More substantively, given the ICML setting, there are no simulations or other forms of evaluation to show the model holds in practice.

**Questions For Authors:**

See above!

**Relation To Broader Scientific Literature:**

This paper offers a novel framing for how GenAI providers might choose to give partial / incomplete / no information in response to key queries, thus incubating discourse and data production on alternative platforms (ensuring additional data is available in the future). The framing is nice, though substantively limited, and as a first step offers several directions for future work to build upon.

**Theoretical Claims:**

I read carefully through 4.1. Selective Response Implies Increased User Proportions and Appendix D.1. They are correct, though the proof is parameter-sensitive and assumes exact retraining each round, no delays or stochasticity. Again, as this is intended as simplified model of the setting this is reasonable. I do, however, believe the paper would benefit from further exposition on the limitations of the work.

---

> ### Author Rebuttal · Authors · 2025-03-28
>
> We thank the reviewer for the useful feedback and for finding our work novel.
>
> We find the works you mentioned, namely Bergemann & Bonatti (2024), McIntyre & Srinivasan (2017), and Shumailov et al. (2023), highly relevant and will incorporate them into our literature review. Thank you for pointing us to them.
>
> Regarding the "*assumptions that introduce some weaknesses to the work*", we fully agree with the reviewer's observation. Our goal was to present a simplified and clean model to highlight the conceptual benefits of using selective responses. While our assumptions impose certain limitations on our results, we have carefully ensured that each assumption is well-motivated. Additionally, we believe that some assumptions could be relaxed while preserving the conceptual contribution, though this will come at the cost of less clean results. We thank the reviewer for bringing this up, as we believe that addressing this point would be a great addition to our paper.
>
> The reviewer mentioned that "*The softmax definition of user choice is symmetric and memoryless… It doesn't account for any of the following effects: (1) User stickiness (2) Frustration penalty if GenAI gives poor/no answers (3) Platform trust*". The reviewer is correct in pointing this out as an aspect we do not address. Capturing all possible factors in a single model is challenging and often makes theoretical analysis impossible. Therefore, these aspects are fascinating research directions that offer opportunities for new stylized theoretical frameworks as well as comprehensive empirical work.

---

> > ### Comment · Reviewer_KDdP · 2025-04-02
> >
> > As long as the authors expand their discussion on limitations (and thus for future work) I will leave my score as it stands.
> >
> > I do think the addition of some form of simulation would strengthen the work given the setting, but leave it to the authors to do as they please. If they agree to add something along these lines I will update my score to the highest accept ✨

---

> > > ### Author Response · Authors · 2025-04-03
> > >
> > > We appreciate the reviewer's quick reply.
> > >
> > > As the reviewer suggested, we intend to expand the discussion on limitations and future work.
> > >
> > > Additionally, we have carefully considered the reviewer’s suggestion regarding simulations. We tried to identify a form of simulation that  "*would strengthen the work given the setting*," which turned out to be a challenging task. After discussion and brainstorming, we believe we have found a way to provide additional insights into the model using simulations, insights that would be difficult to derive formally.
> > >
> > > Below, we outline the simulations we intend to add to the paper. Importantly, ***we consider this to be a very minor change.*** The total runtime is under an hour on a standard laptop, and the corresponding write-up will take just few hours. We can either put this part in the appendix or use the extra page, so no editing is needed. We mention this to reassure the reviewer (and area chair) that this is not a major revision and certainly not an entirely new submission.
> > >
> > > The simulations analyze how profitable it is for GenAI to deviate from the full response strategy, providing a sensitivity analysis of the following parameters:
> > > 1. The discount factor $\gamma$.
> > > 2. The power $\alpha$, where we model the revenue as $r(p)= p^\alpha$.
> > >
> > > Intuitively, selective response becomes superior with greater values of $\gamma$ and $\alpha$. The reasoning is simple: A low discount factor implies a strong preference for the present over the future, making it suboptimal to "waste" present users by not responding. Similarly, a high power $\alpha$ implies that sacrificing present users can result in superlinear revenue gains in the future, making selective response more attractive.
> > >
> > > We use our ASR algorithm to test these intuitions, as well as a staircase heuristic for instances where Assumption 2.1 does not hold. Since both methods are suboptimal, any revenue improvement over the full response strategy implies that the optimal policy would achieve even higher gains.
> > >
> > > Our simulations fully support this intuition.
> > >
> > > We believe these simulations will strengthen the paper, and we are grateful to the reviewer for encouraging this addition. Thank you!

---

### Official Review · Reviewer_TMmF · 2025-03-14

**Overall Recommendation:** 4

**Summary:**

In this paper, the authors introduce a selective response model where a "GenAI system" behaves strategically -- e.g, giving lower quality response, does not respond, etc. They provide a game theoretic based proof that shows that selective response can be beneficial in practice, when the other option is to utilize a human-based forum.

**Claims And Evidence:**

Claims are well-supported by evidence

**Essential References Not Discussed:**

- The authors could connect their work to the selective prediction / deferral literature  (though their model is different from standard prediction-based tasks studied in prior works) -- for example, [1,2]

[1] Hemmer, Patrick, et al. "Learning to defer with limited expert predictions." Proceedings of the AAAI Conference on Artificial Intelligence. Vol. 37. No. 5. 2023.
[2] Mozannar, Hussein, and David Sontag. "Consistent estimators for learning to defer to an expert." International conference on machine learning. PMLR, 2020.

**Experimental Designs Or Analyses:**

- The game theoretical model makes sense, though it debatable how ethical some choices (e.g., providing partial knowledge etc.) is meaningful. Under this case, the two actions that GenAI can take seem limited (either generate or defer).
- The utility of such an approach in a real-world setting (via simulation) would have improved the paper, and clarified the findings

**Methods And Evaluation Criteria:**

The work is primarily theoretical

**Other Comments Or Suggestions:**

- Can authors expand on how models would scale to multiple GenAI agents?

**Other Strengths And Weaknesses:**

- Work addresses a meaningful gap in literature, and clearly elucidates limitations (e.g. assumption of single algorithmic competitor, non-strategic forum behavior, etc.)
- However, the scope of such selective response strategies in the real world could be considered more closely -- for example, is actively withholding knowledge possible?

**Questions For Authors:**

- Can authors expand on how models would scale to multiple GenAI agents?
- Do some of the selective actions -- for example, actively withholding knowledge -- seem feasible in the real world?  That is, do we expect value in any action except choosing not to produce an output vs producing an output?

**Relation To Broader Scientific Literature:**

- The work relates to optimal policies that GenAI should adopt, assume a multi-competition market
- The work is also related to algorithmic deferral [1], which the authors have not considered explicitly in the related work section

References:
[1] Hemmer, Patrick, et al. "Learning to defer with limited expert predictions." Proceedings of the AAAI Conference on Artificial Intelligence. Vol. 37. No. 5. 2023.

**Theoretical Claims:**

Mathematical proofs were not thoroughly checked

---

> ### Author Rebuttal · Authors · 2025-03-28
>
> We appreciate the reviewer's positive and helpful feedback, we hope to leverage it to improve our paper.
>
> Indeed, our work  "*is also related to algorithmic deferral.*" We thank the reviewer for making this connection, which we weren't aware of. We shall address the two papers in our related work.
>
> The reviewer mentions that "*the scope of such selective response strategies in the real world could be considered more closely,*" asking whether "*actively withholding knowledge possible.*"
>
> We believe that selective responses are indeed feasible. To exemplify, recommender systems often face exploration-exploitation tradeoffs, where taking a risky action today (uncertain whether the user will like it or not) trades present rewards with future gains. In this example, the system "selectively" chooses the non-myopic option (sub-optimal in the present) to gain information to improve its utility in the future. In our paper, GenAI prefers inaccurate answers and (slightly) unsatisfied present users with more accurate answers and satisfied future users.
>
> Does this make sense? We came up with this argument while discussing your review. If you find it convincing, we can elaborate on it and include it in the paper.
>
> Answering your specific questions,
>
> 1. Regarding extending on "*how models would scale to multiple GenAI agents.*" Multiple GenAI platforms could induce a Prisoner's Dilemma-like interaction. If one platform chooses to withhold responses, it benefits all other platforms while preserving the same data generation dynamics as always responding. Conversely, if all platforms choose to withhold, they would collectively be better off. In this scenario, the payoffs (expected revenue) depend on the decisions of all platforms as well as user preferences. As in the classic Prisoner's Dilemma, collective cooperation (i.e., not answering) could be enforced through contracts or incentivized by other means. This is a fascinating research direction, and we intend to expand on it in our discussion. Thanks for bringing this up.
>
>
> 2. Regarding "*do we expect value in any action except choosing not to produce an output vs producing an output?*" Just to be sure: Does the reviewer mean answering with varying quality? That is, could it provide lower-quality answers than GenAI could produce deliberately? If that is the case, from a theoretical perspective and for our purposes, there is no substantial difference between providing low-quality answers to all users or providing the highest-quality answers to a subset of users. All the theoretical results extend to lower-quality answers as well. Selective response serves as the mechanism for controlling whether GenAI invests its full efforts with the hope of driving users to generate more data.

---

> > ### Comment · Reviewer_TMmF · 2025-04-07
> >
> > Thanks to the authors for responding! Regarding 2. the question was about the real world implications of withholding knowledge, even when it could have been answered. That is, is actively withholding knowledge an ethical response? I think your response partially addresses this already

---

> > > ### Author Response · Authors · 2025-04-09
> > >
> > > Thank you for your comment.
> > >
> > > One way to think about withholding information (or responding with lower-than-possible quality) is as a means to trade off the utility of current users with the utility of future users. This principle lies at the heart of online algorithms---balancing exploration and exploitation.
> > >
> > > To make this more concrete, consider the Upper Confidence Bound (UCB) algorithm or Thompson Sampling. These algorithms occasionally choose actions (arms) not because they are currently optimal but because they have not been sufficiently explored. As a result, some users receive suboptimal outcomes purely to improve the system's future performance. This is analogous to the case we discuss, where certain users may receive lower-quality answers to improve the quality of information available to future users.
> > >
> > > It's worth noting that one of the original motivations for multi-armed bandits came from medical treatment allocation [1]. Thompson's seminal paper [2], published in Biometrika, laid the foundation for adaptive clinical trials---clearly a domain involving high-risk, high-reward scenarios where such exploration-exploitation tradeoffs are not only present but critical.
> > >
> > > Once again, we emphasize our belief that such actions must be transparent. Specifically, users should be notified that the system employs such a strategy and informed that this is done to improve overall social welfare.

---

### Official Review · Reviewer_E2o5 · 2025-03-20

**Overall Recommendation:** 3

**Summary:**

The paper introduces a novel strategy called "selective response" for Generative AI (GenAI) systems, particularly in the context of human-based forums like Stack Overflow. The main idea is that GenAI could strategically provide inaccurate or conservative responses to queries involving emerging topics and novel technologies, thereby driving users to use human-based forums. This approach aims to create a compounding effect on the data generation process, ultimately increasing both GenAI's revenue and user welfare in the long term. The paper presents a game-theoretic model to explore the dynamics of content creation, welfare, and revenue. Key contributions include:

- Conceptual Contribution: The paper is the first to explore selective response for GenAI, proposing a model where GenAI strategically chooses when, if, and how to engage with user queries.

- Technical Contribution: The authors demonstrate that selective response can Pareto-dominate the always-responding approach, improving user welfare and GenAI's revenue. They also provide an approximately optimal algorithm for maximizing GenAI's revenue under social welfare constraints.

- Regulatory Perspective: The paper derives sufficient and necessary conditions for selective response to improve welfare improvements, offering insights for regulators.

**Claims And Evidence:**

The claims made in the paper are generally supported by clear and convincing evidence. The authors provide a detailed game-theoretic model and derive theoretical results that support their claims. For example, Theorem 1.1 and Theorem 1.2 are supported by rigorous proofs and examples. The paper also includes a discussion on the long-term effects of selective response, supported by Theorem 4.1 and Theorem 4.4. However, some claims, particularly those related to the practical implementation of selective response, could benefit from more empirical validation.

**Essential References Not Discussed:**

The paper does a good job of citing relevant literature, but it could benefit from discussing more recent work on the interaction between AI systems and human content creation platforms. For example, recent studies on the impact of AI on user behavior and data generation in online forums could provide additional context for the proposed selective response strategy.

**Experimental Designs Or Analyses:**

The paper is primarily theoretical, and as such, it does not include experimental designs or analyses. However, the theoretical analyses are sound and well-justified. The authors provide examples and simulations to illustrate their findings, which help to validate the theoretical results.

**Methods And Evaluation Criteria:**

The proposed methods and evaluation criteria are appropriate for the problem at hand. The game-theoretic model is well-suited to analyze the strategic interactions between GenAI and human-based forums. The authors use a sequential setting over discrete rounds, which allows them to explore the dynamics of content creation, welfare, and revenue. The evaluation criteria, such as the accuracy function and the utility functions, are well-defined and align with the objectives of the study.

**Other Comments Or Suggestions:**

The paper is well-written and clearly presents its contributions. However, it could benefit from a more detailed discussion of the limitations and potential challenges of implementing selective response in practice. Additionally, the authors could consider exploring the ethical implications of selectively withholding information from users.

**Other Strengths And Weaknesses:**

Strengths:

- The paper introduces a novel and creative approach (selective response) that has the potential to significantly impact the development of GenAI systems.

- The theoretical analysis is rigorous and well-supported by proofs and examples.

- The paper provides valuable insights for both GenAI companies and regulators, making it relevant to both academic and industry audiences.

Weaknesses:

- The paper is primarily theoretical, and the practical implementation of selective response is not explored in depth. Empirical validation would strengthen the paper's claims.

- The model assumes a single GenAI platform, which may not fully capture the complexities of real-world scenarios with multiple competing platforms.

**Questions For Authors:**

- Question 1: How do you envision the practical implementation of selective response in real-world GenAI systems? What are the potential challenges and how might they be addressed?

Response Impact: A detailed discussion on practical implementation would provide more confidence in the feasibility of the proposed strategy.

- Question 2: Have you considered the ethical implications of selectively withholding information from users? How might this impact user trust and satisfaction?

Response Impact: Addressing ethical concerns would strengthen the paper's relevance and appeal to a broader audience.

- Question 3: Could you elaborate on how the model might be extended to account for multiple competing GenAI platforms?

Response Impact: Extending the model to multiple platforms would make the analysis more robust and applicable to real-world scenarios.

**Relation To Broader Scientific Literature:**

The paper is well-situated within the broader scientific literature on generative AI and game theory. It connects to emerging research on foundation models, competition between generative AI and human content creators, and the impact of generative AI on content diversity. The authors also draw on economic literature on information design and signaling, which adds depth to their analysis. The paper builds on prior work by introducing a novel approach (selective response) and demonstrating its potential benefits.

**Theoretical Claims:**

The theoretical claims in the paper are supported by detailed proofs. For example, Theorem 4.1 is supported by a proof sketch that demonstrates the compounding effect of selective response. Theorem 4.4 is supported by an approximately optimal algorithm (ASR) and its analysis. The proofs are generally clear and convincing, though some technical details are deferred to the appendix, which is common in theoretical papers.

---

> ### Author Rebuttal · Authors · 2025-03-26
>
> We appreciate the reviewer's thoughtful and positive evaluation of our work.
>
> Regarding "*The paper does a good job of citing relevant literature, but it could benefit from discussing more recent work on the interaction between AI systems and human content creation platforms*". Given your advice, we have discovered [1-3] below, which we intend to address in the paper. If you have additional pointers to relevant work, we would appreciate it.
> Addressing the reviewer's specific questions and concerns,
>
> 1. Regarding "*How do you envision the practical implementation of selective response in real-world GenAI systems?*" Platforms like Stack Overflow, providing both GenAI services [4] and user-driven forums, could implement selective response mechanisms. In the broader GenAI market, companies may need to establish contractual agreements governing the use of selective responses. For example, platforms could selectively respond to queries about emerging topics by withholding answers when the LLM of all major companies' confidence falls below a predefined threshold. Given your question, we shall elaborate on this aspect in the paper.
>
>
> 2. Regarding "*Have you considered the ethical implications of selectively withholding information from users? How might this impact user trust and satisfaction?*" Choosing not to answer may be beneficial when GenAI is uncertain about the correctness of its response (or if the response is not sufficiently accurate). In such cases, users may appreciate transparency if they are informed in advance that the expected answer quality is low. We do acknowledge that aggressive selective response could lead to an undesired behavior if this transparency is perceived as unreliability. Indeed, our work suggests opportunities and freedom to apply selective response strategies to real-world systems. We shall address this in our discussion.
>
>
> 3. The reviewer asks, "*Could you elaborate on how the model might be extended to account for multiple competing GenAI platforms?*" Multiple GenAI platforms could induce a Prisoner's Dilemma-like interaction. If one platform chooses to withhold responses, it benefits all other platforms while preserving the same data generation dynamics as always responding. Conversely, if all platforms choose to withhold, they would collectively be better off. In this scenario, the payoffs (expected revenue) depend on the decisions of all platforms as well as user preferences. As in the classic Prisoner's Dilemma, collective cooperation (i.e., not answering) could be enforced through contracts or incentivized by other means. This is a fascinating research direction, and we intend to expand on it in our discussion, thanks for bringing this up.
>
> References:
>
> [1] R Maria del Rio-Chanona, Nadzeya Laurentsyeva, Johannes Wachs, Large language models reduce public knowledge sharing on online Q&A platforms.
>
> [2] Gordon Burtch, Dokyun Lee, Zhichen Chen, The consequences of generative AI for online knowledge communities.
>
> [3] Xinyu Li, Keongtae Kim, Impacts of generative AI on user contributions: evidence from a coding Q&A platform.
>
> [4] https://stackoverflow.blog/2023/07/27/announcing-overflowai/

---

### Decision · Program_Chairs · 2025-05-01

**Decision:**

Accept (poster)

**Comment:**

This manuscript received favourable reviews from all 4 reviewers. The reviews and discussions highlighted both strengths and weaknesses but the general tendency is in favour.

Among the strengths, it was commented that the paper introduces a novel and creative approach (selective response) that has the potential to significantly impact the development of GenAI systems, with rigorous analyses and well-supported by proofs and examples, that the work addresses a meaningful gap in literature, and clearly elucidates limitations (e.g. assumption of single algorithmic competitor, non-strategic forum behavior, etc.), the setting is novel and of interest to the community, and the paper is generally well-written.

Among the weaknesses, it was commented that the assumptions could weaken the work, given the ICML setting, there are no simulations or other forms of evaluation to show the model holds in practice, whereas the paper is primarily theoretical, and the practical implementation of selective response is not explored in depth. Empirical validation would strengthen the paper's claims. Also that the scope of such selective response strategies in the real world could be considered more closely.

The rebuttals alleviated some concerns and the reviewers reiterated their positive evaluation of this paper.